# Normalization Equivariance for Arbitrary Backbones, with Application to Image Denoising

**Youssef Saied** [1]  **François Fleuret** [1] [2]

## Abstract

Normalization Equivariance (NE) is a structural prior that improves robustness to distribution shift in image-to-image tasks. A function $f$ is normalization equivariant iff $f(ay + b\mathbf{1}) = af(y) + b\mathbf{1}$ for all $a > 0$ and $b \in \mathbb{R}$. Existing NE methods constrain every internal layer to NE-compatible operations. These constraints add runtime cost and exclude standard transformer components such as softmax attention and LayerNorm. We introduce Wrapped Normalization Equivariance (WNE), a parameter-free wrapper that normalizes the input, applies any backbone, and denormalizes the output. We prove every NE function admits this factorization, so the wrapper exactly parameterizes the class of NE functions. On blind denoising, wrapping CNN and transformer architectures improves robustness under noise-level mismatch with no measurable GPU overhead, while architectural NE baselines are up to $1.6\times$ slower.

## 1. Introduction

Equivariance priors improve robustness to distribution shift in denoising and related image-to-image tasks. Scale equivariance (SE) and the strictly stronger normalization equivariance (NE) are properties shared by many classical denoisers (Rudin et al., 1992; Buades et al., 2005; Dabov et al., 2007; Gu et al., 2014):

$$
\begin{aligned}
\text{SE:} \quad & f(ay) = af(y) && \text{for } a > 0. && (1) \\
\text{NE:} \quad & f(ay + b\mathbf{1}) = af(y) + b\mathbf{1} && \text{for } a > 0,\ b \in \mathbb{R}. && (2)
\end{aligned}
$$

These equivariances were recently adopted to improve neural network robustness in blind denoising, where the noise

---
[1]Department of Computer Science, University of Geneva, Switzerland [2]Meta FAIR. Correspondence to: Youssef Saied <youssef.saied@unige.ch>.

*Proceedings of the 43rd International Conference on Machine Learning*, Seoul, South Korea. PMLR 306, 2026. Copyright 2026 by the author(s).

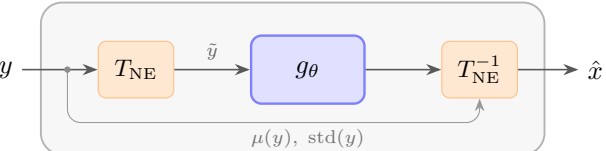

*Figure 1.* **Normalization-equivariant wrapper (WNE).** Given a noisy input $y$, we compute instance statistics $\mu(y)$ and $\text{std}(y)$, normalize to $\tilde{y} = T_{\text{NE}}(y)$, apply an arbitrary backbone $g_\theta$, and denormalize with the stored statistics to produce $\hat{x} = \text{std}(y)\, g_\theta(\tilde{y}) + \mu(y)\mathbf{1}$. This enforces input-output normalization equivariance without modifying the backbone.

level is unknown at test time and may differ from training (Mohan et al., 2020; Herbreteau et al., 2023). This robustness has since motivated applications beyond denoising, including inverse problems and diffusion-style pipelines (Kadkhodaie & Simoncelli, 2021; Hong et al., 2024; Guth et al., 2025; Levac et al., 2025).

Existing methods enforce SE in neural networks by removing biases and leaving only positively 1-homogeneous operations (Mohan et al., 2020). Enforcing NE additionally requires preserving the affine action $y \mapsto ay + b\mathbf{1}$ through every layer, e.g., via affine-constrained convolutions, special nonlinearities, and affine residuals (Herbreteau et al., 2023). Both approaches rule out standard transformer components—softmax and LayerNorm do not commute with $y \mapsto ay + b\mathbf{1}$. The NE-specific layers also add runtime cost (Table 2).

We show that internal-layer constraints are unnecessary. Every NE function admits a normalize-process-denormalize factorization (Characterization 1). Using the characterization, we construct a parameter-free wrapper (WNE) that enforces input-output NE around any backbone without modifying it, including transformers (Figure 1). In blind denoising under noise-level mismatch, WNE matches architectural NE trends on DnCNN. WNE also brings the same robustness pattern to SwinIR and Restormer. In our GPU benchmarks, WNE adds no measured overhead.

**Our contributions:**

- **Method:** parameter-free drop-in wrapper enforcing NE on any backbone, including transformers (Section 3.1).

- **Theory:** characterization showing that normalize-process-denormalize completely parameterizes the NE function class (Section 3.2).

- **Experiments and analysis:** mismatch-robustness gains across CNN and transformer denoisers with negligible runtime cost; a normalized-coordinate analysis explains the trend (Sections 4–5).

**Code:** The implementation is available in the repository `YoussefSaied/normalization_equivariance`.

## 2. Related Work

**Equivariance priors: scale and normalization equivariance.** Two related equivariance priors appear in denoising and related tasks.

Scale equivariance (SE), or positive 1-homogeneity, requires $f(ay) = af(y)$ for $a > 0$. Mohan et al. (2020) propose SE bias-free CNN denoisers, inspiring similar bias-free designs in later work (Zhang et al., 2022; Zamir et al., 2022). Scale equivariant denoisers also serve as implicit priors in inverse problems (Kadkhodaie & Simoncelli, 2021) and appear in energy-based probability density and diffusion models (Guth et al., 2025).

Normalization equivariance (NE) is strictly stronger than SE, requiring $f(ay + b\mathbf{1}) = af(y) + b\mathbf{1}$ for $a > 0, b \in \mathbb{R}$. Herbreteau & Kervrann (2024) survey NE properties across supervised and self-supervised methods, noting that some classical denoisers (e.g., TV, non-local means) are NE (Rudin et al., 1992; Buades et al., 2005) while others such as BM3D (Dabov et al., 2007) and WNNM (Gu et al., 2014) satisfy SE but not NE. Herbreteau et al. (2023) introduce NE as a robustness prior for denoising, and NE predictors have also been leveraged in plug-and-play MRI reconstruction (Hong et al., 2024) and diffusion-based posterior samplers learned from noisy measurements (Levac et al., 2025).

**Enforcing normalization equivariance.** For clarity, we describe three levels at which NE can be enforced.

ARCHITECTURE-LEVEL: Herbreteau et al. (2023) constrain every internal operation to NE-compatible families (affine-constrained convolutions, sorting nonlinearities, affine residuals); this is incompatible with attention and LayerNorm.

OBJECTIVE-LEVEL: Levac et al. (2025) embed an approximate NE property into a SURE-based loss to learn diffusion models from noisy measurements alone (no clean targets), enabling denoiser generalization below the measurement noise level. Their setting is self-supervised and complementary to ours.

INPUT-OUTPUT-LEVEL: (this work) we enforce NE analytically around an arbitrary backbone. Our if and only if characterization (Characterization 1) shows that normalize-process-denormalize reparameterizes the full NE function class rather than restricting it.

**Canonicalization and normalize-process-denormalize.** A general recipe for equivariant maps is to canonicalize the input to a group-orbit representative, apply an arbitrary function, and map back (Kaba et al., 2023; Mondal et al., 2023). In our setting, the orbit representative is obtained by removing the global brightness and contrast degrees of freedom. Characterization 1 shows that this construction parameterizes the full NE function class for the affine action $y \mapsto ay + b\mathbf{1}$. Reversible Instance Normalization (RevIN) uses a related normalize-process-denormalize template, motivated empirically by distribution shift in time-series forecasting (Kim et al., 2022; Liu et al., 2022). We show that the same template, with global $\mu$ and $\mathrm{std}$ statistics, is exactly the form taken by every normalization-equivariant map (Characterization 1).

**Single-noise blind denoising.** We use blind denoising as a controlled stress test for robustness to unknown corruption strength. Following Mohan et al. (2020); Herbreteau et al. (2023), we train at a single $\sigma_{\mathrm{train}}$ and test at $\sigma_{\mathrm{test}} \neq \sigma_{\mathrm{train}}$, probing out-of-distribution generalization rather than interpolation within a trained $\sigma$-range. The same robustness matters in plug-and-play reconstruction, where $\sigma$ controls the denoising strength (Zhang et al., 2022) and is usually a tuned hyperparameter. In dynamic preconditioned PnP this is sharpened: the iteration-dependent preconditioner means the right $\sigma_k$ changes every iteration and is not known a priori. Hong et al. (2024) use an NE denoiser, which allows a single network to absorb this variation, avoiding per-iteration $\sigma$-tuning.

## 3. Method

**Setup and notation.** An instance is the tensor passed to the backbone: during training we use patches and at test time we use full images. For an image of size $C \times H \times W$, we view an instance $y$ as a vector in $\mathbb{R}^d$ with $d = CHW$. All definitions below apply per instance for arbitrary spatial sizes $H, W$. Let $\mathbf{1} \in \mathbb{R}^d$ denote the all-ones instance. In all our experiments, backbones are size-agnostic, so $g_\theta$ is defined for any $H, W$.

**Pooling domain.** All statistics below are computed jointly over all entries (all channels, all pixels). This choice matches the NE group action $y \mapsto ay + b\mathbf{1}$, which acts identically on all entries. Pooling statistics per channel would instead correspond to enforcing a larger, channel-wise affine equivariance, which we do not assume. Thus,

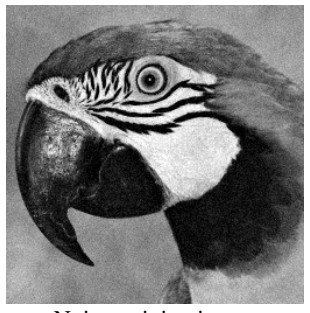
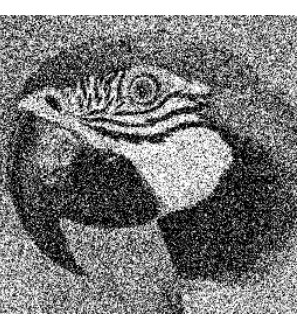
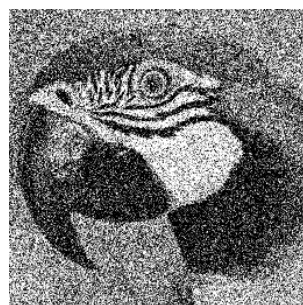
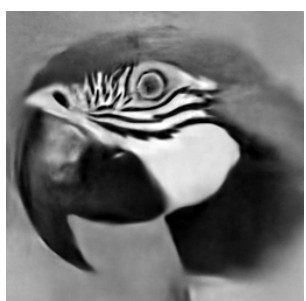

| Noisy training image $\sigma = 10$ | Noisy test image $\sigma = 90$ | Test image denoised by SwinIR | Test image denoised by SwinIR-WNE (ours) |

*Figure 2.* **SwinIR qualitative denoising under noise-level mismatch.** Models are trained at $\sigma_{\text{train}} = 10$ and evaluated at $\sigma_{\text{test}} = 90$, far beyond the training noise. The unwrapped Baseline SwinIR degrades severely, while *SwinIR-WNE* remains stable.

the pooling domain is part of the equivariance specification, not an implementation convention.

For any single instance $y \in \mathbb{R}^d$, define the empirical mean and standard deviation:

$$\mu(y) := \frac{1}{d}\sum_{i=1}^{d} y_i, \qquad \text{std}(y) := \frac{1}{\sqrt{d}}\|y - \mu(y)\mathbf{1}\|_2. \quad (3)$$

We also define the normalized manifold:

$$\mathcal{M} := \{z \in \mathbb{R}^d : \mu(z) = 0,\ \text{std}(z) = 1\} \quad (4)$$
$$= \{z \in \text{Span}(\mathbf{1})^{\perp} : \|z\|_2 = \sqrt{d}\}. \quad (5)$$

### 3.1. Normalization-Equivariant Wrapper

We define the NE normalization map (with a constant-instance guardrail)

$$T_{\text{NE}}(y) := \begin{cases} \frac{y - \mu(y)\mathbf{1}}{\text{std}(y)} & \text{if } \text{std}(y) > 0, \\ 0 & \text{if } \text{std}(y) = 0, \end{cases} \quad (6)$$

so that $T_{\text{NE}}(y) \in \mathcal{M}$ whenever $\text{std}(y) > 0$. Given any backbone $g_\theta : \mathbb{R}^d \to \mathbb{R}^d$, our NE-wrapped map WNE is

$$f_\theta^{\text{WNE}}(y) := \text{std}(y)\, g_\theta\big(T_{\text{NE}}(y)\big) + \mu(y)\mathbf{1}. \quad (7)$$

By construction, $f_\theta^{\text{WNE}}$ is normalization-equivariant: for all $a > 0$ and $b \in \mathbb{R}$,

$$f_\theta^{\text{WNE}}(ay + b\mathbf{1}) = a\, f_\theta^{\text{WNE}}(y) + b\mathbf{1}, \quad (8)$$

with a short proof in Appendix C.3. This guarantee is independent of the backbone internals because the wrapper removes and re-injects the global shift and scale via $\mu(y)$ and $\text{std}(y)$, so $g_\theta$ is only queried on normalized inputs $T_{\text{NE}}(y) \in \mathcal{M}$ (when $\text{std}(y) > 0$). In particular, $g_\theta$ may use components incompatible with internal-layer NE—softmax attention, LayerNorm, and BatchNorm—without breaking the equivariance identity. An equivalent residual-prediction formulation is given in Appendix F.

**Numerical stability.** Our theory assumes the ideal wrapper with a guardrail for constant inputs ($\text{std}(y) = 0$). For numerical stability, the implementation uses $\text{std}_\varepsilon(y) := \text{std}(y) + \varepsilon$ ($\varepsilon = 10^{-5}$). The resulting map is approximately NE: it matches the ideal wrapper whenever $\text{std}(y)$ is not close to 0, and differs only in the near-constant regime where stabilization is active.

**Training objective.** We train by minimizing raw-space MSE:

$$\min_\theta\ \mathbb{E}\big[\|f_\theta^{\text{WNE}}(y) - x\|_2^2\big]. \quad (9)$$

**Relationship to Instance Normalization.** The normalization $T_{\text{NE}}(y) = (y - \mu(y)\mathbf{1})/\text{std}(y)$ has the same algebraic form as instance normalization when statistics are pooled over all channels and pixels (as opposed to the common per-channel variant). The key difference is that the wrapper reuses the input instance statistics to analytically invert the transform at the output (7).

### 3.2. Characterization of NE Maps

Our method and analysis rely on the following complete characterization of normalization-equivariant maps.

**Characterization 1** (Characterization of NE maps). *A function $f : \mathbb{R}^d \to \mathbb{R}^d$ is normalization-equivariant if and only if there exists a function $g : \mathcal{M} \to \mathbb{R}^d$ such that, for all $y$ with $\text{std}(y) > 0$,*

$$f(y) = \text{std}(y)\, g\big(T_{\text{NE}}(y)\big) + \mu(y)\mathbf{1}.$$

*Moreover, for $\text{std}(y) = 0$ one necessarily has $f(y) = \mu(y)\mathbf{1}$, and $g$ is uniquely determined by $f$ on $\mathcal{M}$.*

Proofs are in Appendix C. (Equivalently, an NE map is determined by its restriction to $\mathcal{M}$, together with the forced behavior on constant inputs.)

*Table 1.* **Matched-noise PSNR (dB).** Average PSNR on Set12 and BSD68 for AWGN with $\sigma \in \{10, 25, 50\}$ (8-bit units). Each model is trained at a single $\sigma$ and evaluated at the same $\sigma$. Our WNE remains competitive while providing robustness under mismatch (Figures 3–4). *SE-arch* and *NE-arch* are FDnCNN-based architectural variants (no BatchNorm, constrained layers), while *WNE* wraps the standard backbone.

| Noise level $\sigma$ (8-bit) | | Set12 | | | BSD68 | | |
|---|---|---|---|---|---|---|---|
| | | 10 | 25 | 50 | 10 | 25 | 50 |
| DnCNN family (Zhang et al., 2017) | *Baseline* | 34.92 | 30.50 | 27.17 | 33.92 | 29.19 | 26.18 |
| | *SE-arch* | 34.41 | 30.42 | 27.19 | 33.51 | 29.12 | 26.12 |
| | *NE-arch* | 34.65 | 30.37 | 27.20 | 33.72 | 29.11 | 26.18 |
| | *WNE* (ours) | 34.77 | 30.43 | 27.22 | 33.80 | 29.15 | 26.20 |
| SwinIR (Liang et al., 2021) | *Baseline* | 35.08 | 30.79 | 27.61 | 34.01 | 29.38 | 26.46 |
| | *WNE* (ours) | 34.95 | 30.65 | 27.47 | 33.94 | 29.30 | 26.37 |

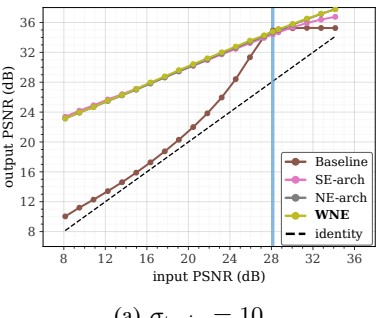 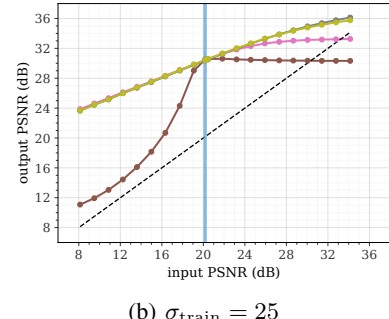 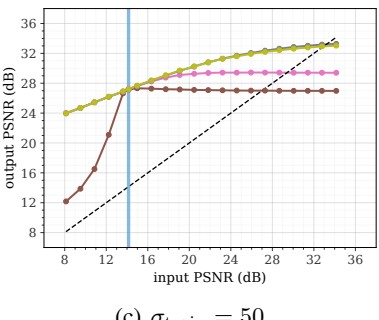

(a) $\sigma_{\text{train}} = 10$        (b) $\sigma_{\text{train}} = 25$        (c) $\sigma_{\text{train}} = 50$

*Figure 3.* Output PSNR versus input PSNR on Set12 for models trained at a single $\sigma_{\text{train}}$ (vertical reference line) and tested at varying $\sigma_{\text{test}}$, following Herbreteau et al. (2023). The Baseline is noise-level specific, while equivariance stabilizes performance under mismatch. Our *WNE* closely matches the mismatch trend of the architectural normalization-equivariant reference (NE-arch, FDnCNN-based), and improves over the architectural scale-equivariant reference (SE-arch) and the unwrapped baseline. Dashed line: no denoising ($\hat{x} = y$).

## 4. Experimental Results

We test whether input-output NE enforcement improves generalization away from the training noise level under a single-noise mismatch diagnostic.

**Protocol and metric.** We follow the single-noise diagnostic of Herbreteau et al. (2023), the single-point limit of the limited-range training used to study mismatch in Mohan et al. (2020). Each model is trained on additive white Gaussian noise (AWGN) at a fixed $\sigma_{\text{train}} \in \{10, 25, 50\}$ (8-bit units) and evaluated on Set12 over a wide range of $\sigma_{\text{test}}$. For each test image, we compute input PSNR (noisy $y$ vs. clean $x$) and output PSNR (denoised $\hat{x}$ vs. $x$). We then plot output PSNR versus input PSNR (Figures 3 and 4). The vertical line indicates the training noise level: moving away from the vertical line corresponds to testing outside the training noise range. The dashed identity line corresponds to leaving the input unchanged.

**Models and compared variants.** We evaluate two widely used reference backbones from distinct families: DnCNN (Zhang et al., 2017) (CNN) and SwinIR (Liang et al., 2021) (transformer-based restoration). We focus on

mismatch trends rather than peak PSNR. For each family, we compare the unmodified backbone (Baseline) to its wrapped counterpart (WNE). Training details and implementation in Appendix A and Appendix B. Additional diagnostics in the appendix cover input-only normalization versus the full wrapper (Appendix O), SSIM diagnostics (Appendix J), soft-NE and multi-noise controls with explicit equivariance-defect measurements (Appendices D–E), and robustness beyond AWGN (Appendix P). Breadth checks extend the evaluation to FDnCNN-family controls (Appendix H), Restormer and color denoising (Appendices K–L), real sensor noise on SIDD (Appendix M), and a Noise2Noise setting where denoisers are reused for iterative sampling and inverse-problem solving (Appendix N).

**DnCNN family (CNN-based).** We compare (i) Baseline, the standard DnCNN implementation (with BatchNorm), and (ii) WNE, our wrapper applied to the same DnCNN backbone. We also report the architectural equivariant references (iii) SE-arch and (iv) NE-arch from Herbreteau et al. (2023), built on FDnCNN with layer replacements throughout (Appendix A.2). These are architectural references rather than controlled same-backbone ablations: relative to standard DnCNN, they remove BatchNorm and bias, and

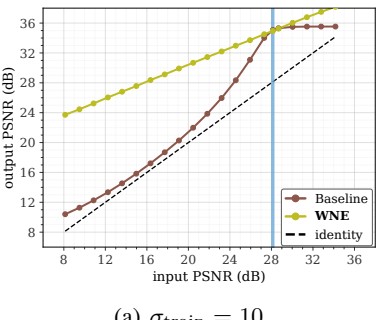 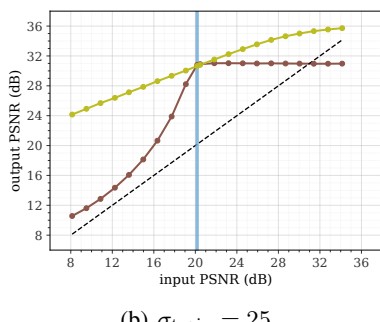 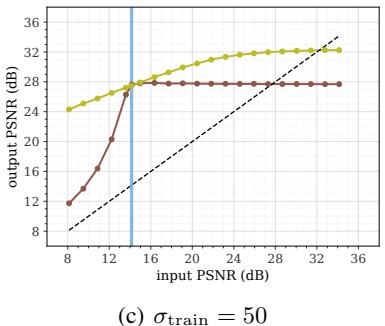

(a) $\sigma_{\text{train}} = 10$    (b) $\sigma_{\text{train}} = 25$    (c) $\sigma_{\text{train}} = 50$

*Figure 4.* **Noise-level mismatch on SwinIR.** Same diagnostic as Figure 3. The Baseline SwinIR degrades once $\sigma_{\text{test}} \neq \sigma_{\text{train}}$, while *SwinIR-WNE* stabilizes performance away from the training noise. Dashed line: no denoising ($\hat{x} = y$).

NE-arch also constrains the admissible filter class. Appendix H brackets this confound with an FDnCNN-family comparison, where WNE-FDnCNN still tracks NE-arch under mismatch.

**SwinIR (transformer-based).** For SwinIR, we compare the Baseline and WNE. We include SwinIR as a modern transformer restoration backbone to test whether the wrapper improves mismatch robustness beyond CNN architectures. Appendix K shows the same mismatch-stabilization pattern on Restormer, a second transformer backbone, and Appendix L extends the same construction to color denoising. For transformers, architectural SE/NE would require modifying attention/LayerNorm; we therefore compare Baseline vs. WNE.

### 4.1. DnCNN: Input-Output NE Matches Architectural NE Trends

Across all three training settings $\sigma_{\text{train}} \in \{10, 25, 50\}$, Figure 3 shows that WNE produces mismatch curves that closely match the robustness trend of the architectural normalization-equivariant NE-arch reference. This indicates that enforcing NE at the input-output level recovers essentially the same robustness as architectural NE without constraining internal layers. Relative to SE-arch, we observe an asymmetric pattern: when training at higher noise and testing at lower noise (high-to-low), WNE improves over SE-arch; when training at lower noise and testing at higher noise (low-to-high), WNE is comparable to SE-arch. Overall, the additional shift-equivariance in NE yields consistent gains over SE in the high-to-low regime.

### 4.2. SwinIR: Improved Extrapolation for a Transformer Backbone

Figure 4 shows that SwinIR exhibits pronounced noise-level specificity under single-noise training: the baseline curve flattens once $\sigma_{\text{test}} \neq \sigma_{\text{train}}$, indicating poor extrapolation outside the training noise. Wrapping SwinIR with WNE stabilizes the mapping, yielding a smoother and more monotonic output-PSNR trend across low and high input noise.

Figure 2 illustrates this effect in a severe mismatch setting: models trained at $\sigma_{\text{train}} = 10$ are evaluated at a much higher noise level ($\sigma_{\text{test}} = 90$). The Baseline SwinIR degrades visibly, while the wrapped model remains qualitatively stable, consistent with the trend in Figure 4.

### 4.3. The Wrapper Does Not Degrade Performance

At matched noise levels ($\sigma_{\text{test}} = \sigma_{\text{train}}$), WNE remains close to the unwrapped baseline for both backbones (Table 1). We report results at the final training checkpoint for all models (no early stopping) to keep evaluation consistent across variants; our primary focus is robustness under mismatch (Figures 3 and 4) rather than peak matched-noise PSNR.

### 4.4. The Wrapper Has Negligible Overhead on GPU

Table 2 reports wall-clock seconds per batch for FDnCNN variants. On GPU, the WNE wrapper adds negligible overhead relative to the baseline, since it only introduces per-instance reductions ($\mu, \text{std}$) and elementwise affine operations around the backbone. The architectural NE reference NE-arch is slower in our benchmark, consistent with the cost of enforcing equivariance throughout internal layers rather than via an outer analytic transform. Similar findings were also reported by Herbreteau et al. (2023). We observe the same behavior for SwinIR: on GPU, WNE matches the baseline within measurement noise ($1.00\times$ backward, $1.00\times$ inference; Appendix Table 5). Full GPU/CPU timings and the benchmarking protocol are reported in Appendix B.1.

### 4.5. Adaptive-Filter Interpretation Extends to Transformers

Figure 5 visualizes two Jacobian rows $(J_f(y))_{i,:}$ of the denoiser $f$ as spatial filters (reshaped to the input grid), following Mohan et al. (2020); Herbreteau et al. (2023).

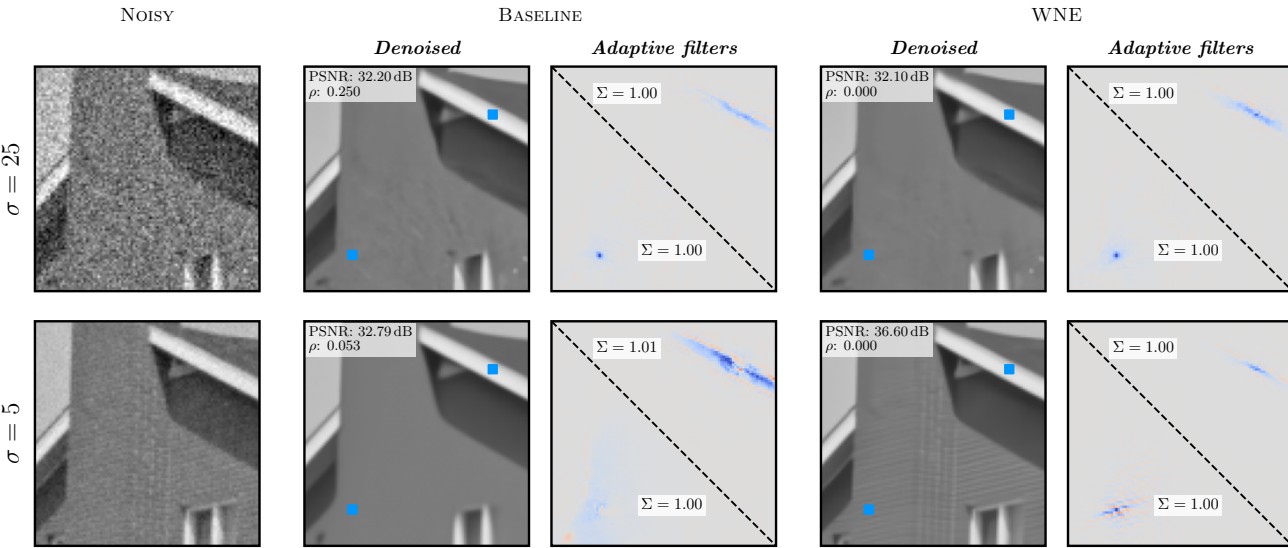

*Figure 5.* **Input-adaptive (affine) filters from the Jacobian.** We visualize two rows of $J_f(y)$ as per-output-pixel filters. For *WNE*, NE implies positive 1-homogeneity, so at differentiable points $f(y) = J_f(y)\, y$ and the filters can be read literally (Mohan et al., 2020; Herbreteau et al., 2023). Under mismatch ($\sigma = 5$), the Baseline SwinIR shows diffuse filters and oversmoothing, while SwinIR-*WNE* retains structured, edge-aligned filters and improves PSNR. Both networks were trained for Gaussian noise at noise level $\sigma = 25$ exclusively.

*Table 2.* **Runtime overhead for FDnCNN on GPU (seconds per batch).** Timing on inputs of shape $16 \times 1 \times 128 \times 128$ with warmup $= 3$ and timed steps $= 10$. "Backward" measures one training iteration (forward + loss + backward), and "Inference" measures forward-only. Values in parentheses are ratios relative to the baseline. Hardware: `NVIDIA GeForce RTX 4090`.

| Variant | Backward (s) | Inference (s) |
|---|---|---|
| *Baseline* | 0.034 | 0.011 |
| *NE-arch* | 0.054 (1.60×) | 0.019 (1.69×) |
| *WNE* | 0.033 (0.99×) | 0.011 (1.00×) |

If $f$ is positively 1-homogeneous and differentiable at $y$, Euler's theorem gives

$$f(y) = J_f(y)\, y, \qquad (10)$$

so each output pixel $f(y)_i$ is exactly the inner product of $y$ with its local filter $(J_f(y))_{i,:}$, making these rows literal input-adaptive filters. Because WNE enforces NE (hence SE), this interpretation applies to SwinIR despite attention and LayerNorm. Moreover, NE implies shift equivariance, and at differentiable points this yields

$$J_f(y)\, \mathbf{1} = \mathbf{1}, \qquad (11)$$

so each filter has unit sum and the local model is an adaptive affine combination of input pixels. To verify the Euler identity in the displayed examples, we report

$$\rho(y) := \frac{\|f(y) - J_f(y)\, y\|_2}{\|f(y)\|_2}, \qquad (12)$$

which is 0 for differentiable positively 1-homogeneous maps (up to numerical error). More broadly, since the wrapper enforces SE/NE at the input-output level independently of the backbone, Jacobian-based analyses developed for bias-free SE denoisers (Mohan et al., 2020; Herbreteau et al., 2023) extend directly to any wrapped architecture. This includes recent generalization analyses based on harmonic structure (Kadkhodaie et al., 2024).

## 5. Analysis

We analyze a mechanism that explains the noise-level mismatch robustness of normalization-equivariant (NE) denoisers. The analysis applies to any NE denoiser (architectural, classical, or wrapped). We focus on (i) the stability of output PSNR under noise-level mismatch and (ii) why performance degrades differently when testing above versus below the training noise level.

All statistics reported in this section were computed using training patches (using the same training corpus as Section 4), corrupted with AWGN at the stated $\sigma$.

### 5.1. NE Reduces Denoising to Normalized Regression

By Characterization 1, any NE map $f : \mathbb{R}^d \to \mathbb{R}^d$ is entirely determined by a function $g : \mathcal{M} \to \mathbb{R}^d$ via

$$f(y) = \text{std}(y)\, g\big(T_{\text{NE}}(y)\big) + \mu(y)\mathbf{1}. \qquad (13)$$

The normalization and denormalization steps are fixed analytic operations; only the map $g$ acting on normalized inputs

$\tilde{y} \in \mathcal{M}$ varies across NE maps.

This factorization makes the source of robustness (or failure) transparent: cross-noise behavior is governed by how $g$ performs on normalized pairs $(\tilde{y}, \tilde{x})$ as $\sigma$ varies.

**PSNR decomposes into normalized and scale terms.** A direct consequence of the factorization is that PSNR decomposes into a normalized error term and a scale term.

For a clean target $x$ and noisy observation $y$, define the matched normalized target:

$$\tilde{x} := T_{\mathrm{NE}}(x\,;\,y) := \frac{x - \mu(y)\mathbf{1}}{\mathrm{std}(y)}. \tag{14}$$

Then (Appendix C.5):

$$f(y) - x = \mathrm{std}(y)\big(g(\tilde{y}) - \tilde{x}\big), \tag{15}$$

so the raw-space MSE factorizes as:

$$\|f(y) - x\|_2^2 = \mathrm{std}(y)^2 \, \|g(\tilde{y}) - \tilde{x}\|_2^2 \tag{16}$$

Thus raw-space MSE is equivalent to a weighted regression in normalized coordinates, with weight $\mathrm{std}(y)^2$ per instance. Consequently, the raw-space PSNR decomposes as:

$$\mathrm{PSNR} = \underbrace{10\log_{10}(dR^2)}_{\text{constant}}$$
$$- \underbrace{10\log_{10}\big(\|g(\tilde{y}) - \tilde{x}\|_2^2\big)}_{\text{normalized error } Q(\tilde{y}, \tilde{x})}$$
$$- \underbrace{20\log_{10}\big(\mathrm{std}(y)\big)}_{\text{scale}}, \tag{17}$$

where $R$ denotes the pixel dynamic range[1]. The scale term depends only on the instance statistic $\mathrm{std}(y)$, while the normalized error

$$Q(\tilde{y}, \tilde{x}) := -10\log_{10}\|g(\tilde{y}) - \tilde{x}\|_2^2 \tag{18}$$

captures regression quality in normalized coordinates (larger $Q$ means smaller normalized squared error). Since the scale term is independent of the denoiser, it acts as a common input-dependent offset; after accounting for this offset, all denoiser-dependent robustness and mismatch behavior is carried by $Q$.

## 5.2. The Geometry of Normalized Coordinates

In normalized coordinates, the input $\tilde{y} = T_{\mathrm{NE}}(y)$ always lies on $\mathcal{M}$ (unit standard deviation, zero mean). The target $\tilde{x} = (x - \mu(y)\mathbf{1})/\mathrm{std}(y)$ does not: as noise increases, $\mathrm{std}(y)$ grows, so $\tilde{x}$ shrinks toward the origin. Thus higher noise compresses the target while leaving the input on $\mathcal{M}$.

---

[1] In our experiments images are scaled to $[0, 1]$, so $R = 1$.

We define the input-target distance in normalized coordinates:

$$\delta := \tilde{y} - \tilde{x}, \qquad \Delta := \|\delta\|_2. \tag{19}$$

Under AWGN ($y = x + \sigma n$, $n \sim \mathcal{N}(0, I_d)$),

$$\delta = \frac{y - x}{\mathrm{std}(y)} = \frac{\sigma}{\mathrm{std}(y)}\,n. \tag{20}$$

As $\sigma \to \infty$, $\mathrm{std}(y) \approx \sigma$ and $\Delta$ concentrates near $\sqrt{d}$, while as $\sigma \to 0$, $\Delta \to 0$ (Figure 7).

**SNR controls $\Delta$.** To understand how $\Delta$ varies with noise level, we relate it to the signal-to-noise ratio. Define the empirical signal variance $\sigma_x^2 := \frac{1}{d}\|x - \mu(x)\mathbf{1}\|_2^2$. For high-dimensional instances, cross-terms concentrate and

$$\mathrm{std}(y)^2 \approx \sigma_x^2 + \sigma^2. \tag{21}$$

Substituting into $\Delta^2 = \|\sigma n/\mathrm{std}(y)\|_2^2$ and taking expectations gives

$$\mathbb{E}[\Delta^2 \mid x, \sigma] \approx \frac{d}{1 + \mathrm{SNR}}, \qquad \mathrm{SNR} := \sigma_x^2/\sigma^2. \tag{22}$$

## 5.3. Normalized Error Is Largely $\Delta$-Driven

**Observation 1** ($\Delta$ predicts normalized MSE). For our model and setup, the conditional normalized MSE satisfies

$$\mathbb{E}\big[\|g(\tilde{y}) - \tilde{x}\|_2^2 \mid \Delta = r,\, \sigma\big] \approx \psi_g(r), \tag{23}$$

with residual dependence on $\sigma$ that is small compared to variation across $r$ (Figure 6).

The intuition is that, after conditioning on $\Delta$, the distribution of normalized pairs $(\tilde{y}, \tilde{x})$ changes little with $\sigma$. Thus the backbone faces a statistically similar normalized regression problem across noise levels. We study why this holds in Appendix G.

To estimate $\mathbb{E}\big[\|g(\tilde{y}) - \tilde{x}\|_2^2 \mid \Delta = r, \sigma\big]$ as a function of $r$, we bin samples by $\Delta$ and compute the mean and standard deviation of $Q(\tilde{y}, \tilde{x})$ within each bin, separately for each $\sigma$. Figure 6 shows that within the central 95% of the training $\Delta$ distribution, normalized error depends mainly on $\Delta$, with little residual dependence on $\sigma$. An extended analysis comparing Baseline and WNE is given in Appendix I. The Baseline exhibits stronger dependence on $\sigma$ at fixed $\Delta$ (Figure 11), suggesting that the collapse is a consequence of enforcing NE rather than an artifact of the difficulty measure itself.

## 5.4. Train-Test Overlap in $\Delta$

**Observation 2** (Substantial overlap across noise levels). The distribution of $\Delta$ shows substantial overlap across noise levels (Figure 7, Table 3).

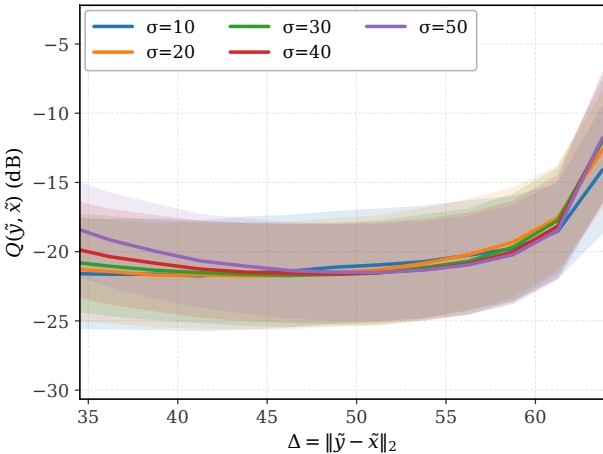

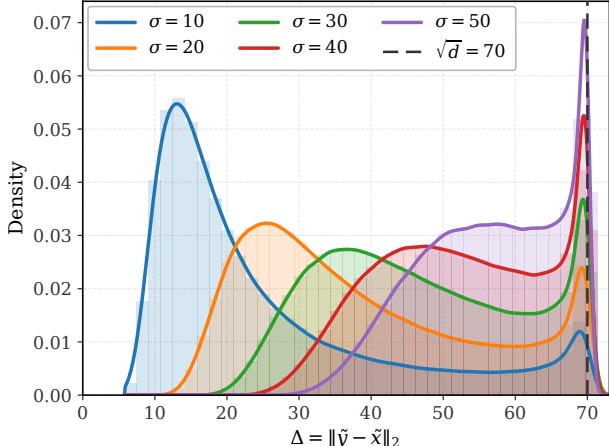

*Figure 6.* **Normalized-space error vs difficulty for NE (central 95% range).** For each $\sigma$ we plot $Q(\tilde{y}, \tilde{x})$ versus $\Delta = \|\tilde{y} - \tilde{x}\|_2$, reporting mean and standard deviation over samples. The $x$-axis shows the central 95% interval of the training $\Delta$ distribution at $\sigma_{\text{train}} = 50$, used as a fixed reference range for all $\sigma$. Within this shared difficulty band, curves for different $\sigma$ show substantial collapse, suggesting that once difficulty is controlled by $\Delta$, the remaining dependence of normalized regression error on $\sigma$ is small. Model: SwinIR-*WNE*, trained at $\sigma_{\text{train}} = 50$.

*Table 3.* **Train-test $\Delta$ overlap for the NE wrapper.** Entries are empirical coverages $m(\sigma_{\text{test}}; \sigma_{\text{train}})$ (percent), measured using the central 95% training interval at each $\sigma_{\text{train}}$. Overlap is substantial but asymmetric: training at low noise covers many higher-noise difficulties, while training at high noise covers only a fraction of very low-noise difficulties. This matches the empirical asymmetry observed in Figure 3.

| $\sigma_{\text{train}} \backslash \sigma_{\text{test}}$ | 50 | 40 | 30 | 20 | 10 |
|---|---|---|---|---|---|
| 50 | 95 | 87 | 68 | 42 | 19 |
| 40 | 97 | 95 | 83 | 56 | 24 |
| 30 | 96 | 96 | 95 | 76 | 34 |
| 20 | 93 | 95 | 96 | 95 | 56 |
| 10 | 88 | 90 | 92 | 95 | 95 |

This overlap reflects the wide variation in $\sigma_x^2$ across natural-image patches (flat regions vs. textured regions). Changing $\sigma$ therefore rescales a broad SNR distribution rather than shifting a point mass, preserving overlap in the induced $\Delta$ values.

To quantify the overlap of $\Delta$ across noise levels, for a given $\sigma_{\text{train}}$ we define the central 95% interval of its $\Delta$ distribution:

$$I(\sigma_{\text{train}}) := \left[ q_{0.025}(\sigma_{\text{train}}), \ q_{0.975}(\sigma_{\text{train}}) \right] \quad (24)$$

where $q_p(\sigma)$ denotes the empirical $p$-quantile of $\Delta$ computed from samples at noise level $\sigma$.

For $\sigma_{\text{test}}$, we empirically estimate the in-range mass as

*Figure 7.* **NE difficulty distributions.** Empirical histograms of $\Delta = \|\tilde{y} - \tilde{x}\|_2$ at several noise levels $\sigma$. As $\sigma$ increases, mass shifts upward and concentrates near $\sqrt{d}$, consistent with $\Delta \approx \|n\|_2$ at high noise. Despite this shift, the distributions still overlap substantially across $\sigma$. Computed from $10^6$ training patches.

follows:

$$m(\sigma_{\text{test}}; \sigma_{\text{train}}) := \frac{1}{N} \sum_{i=1}^{N} \mathbf{1}\{\Delta_{\sigma_{\text{test}}}^i \in I(\sigma_{\text{train}})\} \quad (25)$$

where $N$ is the number of samples.

Table 3 shows that this in-range mass remains substantial off-diagonal, so mismatch test samples often query difficulties that are well-represented during training.

### 5.5. Mechanism for Robustness and Asymmetry

Normalized regression error depends mainly on $\Delta$ rather than on $\sigma$ (Observation 1): an input whose difficulty $\Delta$ was well represented in training is denoised with similar normalized error even when that $\Delta$ now arises from an unseen noise level. Observation 2 (Figure 7) shows that the $\Delta$-distributions can overlap substantially across $\sigma$, so under mismatch $g$ can still denoise when queried at familiar difficulties. Figure 3 shows that generalization across noise levels is asymmetric: a denoiser generalizes to higher test noise more readily than to lower test noise. The central $\Delta$-interval is wide at low $\sigma_{\text{train}}$ but narrower at high $\sigma_{\text{train}}$, so low-noise training covers most higher-$\sigma_{\text{test}}$ difficulties, whereas high-noise training covers progressively fewer lower-$\sigma_{\text{test}}$ ones.

## 6. Conclusion

We prove that a map is normalization equivariant if and only if it factors as normalize, process, denormalize. The "if" direction means we can enforce NE by wrapping any backbone; the "only if" direction means we lose nothing by

doing so, because every NE map is already of this form. Enforcing exact NE therefore leaves the backbone architecture unrestricted.

This gives WNE: wrap any backbone with the analytic normalize-denormalize pair, and the result is exactly NE. No internal layers change. Attention, LayerNorm, and Batch-Norm all stay where they are.

In blind denoising, WNE improves robustness to noise-level mismatch for DnCNN and SwinIR, remains competitive at matched noise, and adds no measured GPU overhead. Restormer (Appendix K), color denoising (Appendix L), and SIDD real-noise results (Appendix M) follow the same pattern.

Our normalized-coordinate view explains why NE improves denoising robustness. An NE denoiser is determined by its action on normalized inputs, so changing $\sigma$ mainly shifts which normalized difficulties $\Delta = \|\tilde{y} - \tilde{x}\|_2$ are queried, not the regression problem at fixed $\Delta$ (Appendix G). When train and test $\Delta$-distributions overlap, the wrapped backbone stays in an interpolation regime. Table 3 shows this overlap is substantial but asymmetric: training at low noise covers many high-noise difficulties, while training at high noise covers fewer low-noise ones. The same asymmetry shows up in the PSNR curves of Figure 3.

The same mechanism extends to iterative reuse, where a sampler queries the denoiser at many noise levels. In Appendix N, a Noise2Noise denoiser trained at $\sigma_{\text{train}} = 10$ is reused unchanged inside a random-inpainting sampler initialized at full Gaussian noise, about $25\times$ above the training scale. With the same backbone, training pairs, and sampler hyperparameters, the wrapped denoiser reaches 23.87 dB while the unwrapped one stays at 6.08 dB.

These results point to natural directions for future work: preconditioned plug-and-play methods that rely on exact NE (Hong et al., 2024), noisy-only diffusion learning (Levac et al., 2025), and broader image restoration tasks.

## Impact Statement

This paper presents a theoretical characterization and a parameter-free wrapper for enforcing normalization equivariance in image-to-image neural networks. The main intended benefit is improved robustness under distribution shift, reducing calibration burden in applications such as photography, medical imaging, and scientific imaging. Potential risks mirror those of stronger image-processing systems in sensitive settings: denoising can alter visual evidence, and benchmark restoration performance alone does not validate real-world deployment. The method releases no personal data or generative model, and we identify no additional misuse risks beyond these application-level concerns.

## Acknowledgments

We thank Marian Lukac, Eleni Kalogirou, and the members of the Machine Learning Group at the University of Geneva for helpful discussions. This work was supported by the Hasler Foundation under the Responsible AI program.

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

# A. Description of the Denoising Architectures and Implementation

## A.1. Description of Models

**DnCNN and FDnCNN:** We implement the Denoising CNN (DnCNN) of Zhang et al. (2017), which consists of 20 convolutional layers with $3 \times 3$ filters and 64 channels, batch normalization (Ioffe & Szegedy, 2015), and ReLU nonlinearities. To construct the bias-free/BatchNorm-free variant used by prior equivariant baselines, we follow Herbreteau et al. (2023) and use FDnCNN, an unpublished flexible DnCNN variant that removes batch normalization and additive bias. In our experiments, we keep the standard DnCNN backbone unchanged when applying our wrapper (WNE), while FDnCNN serves as the base architecture for the architectural scale-equivariant and normalization-equivariant baselines (SE-arch, NE-arch) where internal-layer equivariance is enforced.

**SwinIR:** We use SwinIR (Liang et al., 2021), a transformer-based denoiser built from Swin Transformer blocks (Liu et al., 2021) (windowed self-attention with LayerNorm and MLPs). We use the denoising configuration (no upsampling) and instantiate the lightweight variant in our code: a shallow $3 \times 3$ convolutional feature extractor, followed by 4 Residual Swin Transformer Blocks (RSTBs) with depths $(6, 6, 6, 6)$ (total 24 Swin Transformer blocks, embedding dimension 60, window size 8, 6 heads, MLP ratio 2; patch size 1 with patch-wise LayerNorm), and a $3 \times 3$ convolutional reconstruction head. As architectural SE/NE counterparts are not available as drop-in replacements for standard transformer components (e.g., attention and LayerNorm), for SwinIR we compare only the unmodified backbone (Baseline) and its wrapped version (WNE).

**Restormer:** We use Restormer (Zamir et al., 2022), a transformer-based denoiser built from Multi-DConv Head Transposed Self-Attention (MDTA) and Gated-Dconv Feed-Forward Network (GDFN) blocks. In our code, we instantiate the denoising variant as a 4-level encoder-decoder with an overlapping $3 \times 3$ convolutional input embedding, base width 48, block counts $(4, 6, 6, 8)$ across the encoder/latent levels, attention heads $(1, 2, 4, 8)$, FFN expansion factor 2.66, and 4 refinement blocks at the final resolution. Downsampling and upsampling are implemented with convolution plus PixelUnshuffle/PixelShuffle, and the network predicts the denoised output through a final $3 \times 3$ convolution with a global residual connection from the noisy input. For the grayscale experiments, the model uses single-channel input/output and is trained on $64 \times 64$ patches. As architectural SE/NE counterparts are not available as drop-in replacements for the standard attention and normalization blocks, for Restormer we compare only the unmodified backbone (Baseline) and its wrapped version (WNE).

## A.2. Description of Variants

**Baseline:** The Baseline variant is the unmodified reference implementation of each backbone (DnCNN or SwinIR), including its standard use of biases and (when present) internal normalization layers.

**SE-arch:** For the DnCNN family, the scale-equivariant baseline SE-arch follows the bias-free ReLU construction of Herbreteau et al. (2023) which is based on the construction from Mohan et al. (2020). Concretely, architectural sources of scale breaking (most notably additive biases, and normalization layers that introduce additive shifts) are removed, leaving only positively 1-homogeneous operations. This enforces scale equivariance $f(ay) = af(y)$ for all $a > 0$.

**NE-arch:** The normalization-equivariant baseline NE-arch follows the architectural construction of Herbreteau et al. (2023). In this design, NE is enforced throughout the network by restricting internal operations to NE-compatible families:

(i) Standard convolutions are replaced by affine-constrained convolutions: bias-free convolutions whose weights are constrained so that a constant feature map is mapped to itself,

$$\mathrm{Conv}_w(\mathbf{1}) = \mathbf{1},$$

which for a multi-channel $3 \times 3$ convolution is equivalent to requiring, for each output channel $c'$,

$$\sum_c \sum_{i,j} w_{c',c,i,j} = 1,$$

with reflect padding.

(ii) ReLU nonlinearities are replaced by SortPool channel-wise sorting patterns, implemented via the two-channel operator

$$\text{SortPool}(u, v) = \big(\min(u, v), \max(u, v)\big),$$

applied in a fixed pairing scheme across channels.

(iii) Classical residual sums are replaced by affine residual connections, where two branches $\ell_1$ and $\ell_2$ are combined as

$$(1 - t)\,\ell_1 + t\,\ell_2,$$

with a trainable scalar $t$.

As in Herbreteau et al. (2023), NE-arch is instantiated on FDnCNN (BatchNorm-free), since enforcing NE throughout internal layers is not compatible with BatchNorm.

**Wrapped normalization-equivariant (WNE):** Our method WNE enforces input-output NE without modifying internal operations. Given an arbitrary backbone $g_\theta$, the wrapped predictor is

$$f_\theta^{\text{WNE}}(y) = \text{std}(y)\, g_\theta(T_{\text{NE}}(y)) + \mu(y)\mathbf{1},$$

as defined in (7). This yields NE at the input-output level for any backbone $g_\theta$, including off-the-shelf DnCNN (with BatchNorm) and SwinIR.

### A.3. Practical Implementation Notes

Our primary comparisons are within-backbone: Baseline versus WNE for DnCNN and SwinIR, isolating the effect of the outer wrapper on a fixed backbone. For the DnCNN family we also report SE-arch and NE-arch from Herbreteau et al. (2023) as architectural references. These models enforce SE/NE by replacing core operations throughout the network (e.g., constrained convolutions and equivariant nonlinearities), so they constitute a distinct architecture rather than a drop-in modification of standard DnCNN. We include them to place WNE in context relative to established architectural-constraint designs.

**Channel-wise sort pooling (NE-arch).** We implement the channel-wise sorting nonlinearity of Herbreteau et al. (2023) using standard pooling primitives. Given a pairing of channels, we compute the per-location maxima and minima for each pair (pooling with kernel size 2 along the channel axis) and concatenate the results. Between successive sort stages, we apply the same fixed channel reindexing as in Herbreteau et al. (2023) to define the next pairing pattern. Since CNN feature channels have no intrinsic ordering, the precise permutation is not semantically meaningful; it simply realizes the prescribed NE-compatible nonlinearity efficiently.

**Affine-constrained convolutions (NE-arch).** For the architectural NE baseline (NE-arch), we use the public implementation released by Herbreteau et al. (2023), including their stable reparameterization of affine-constrained convolution kernels. (We do not re-derive or redesign these constrained layers; we reuse their tested implementation to keep the architectural NE baseline faithful.)

**Numerical stability (WNE).** For the wrapper, we compute $\mu(y)$ and $\text{std}(y)$ jointly over all entries (all channels and pixels), matching the NE group action $y \mapsto ay + b\mathbf{1}$. All theoretical results (and Proposition 2) assume the idealized wrapper using $\text{std}(y)$ together with the constant-instance guardrail $T_{\text{NE}}(y) = 0$ when $\text{std}(y) = 0$. In code, we use $\text{std}_\varepsilon(y) := \text{std}(y) + \varepsilon$ with $\varepsilon = 10^{-5}$ to stabilize normalization when $\text{std}(y)$ is very small. This yields an approximately NE implementation: it matches the theoretical wrapper whenever $\text{std}(y)$ is not close to $0$, and differs only in the near-constant regime where the stabilization is active.

**Code.** All models are implemented in PyTorch (Paszke et al., 2019). For NE-arch, we rely on the released code of Herbreteau et al. (2023). Our wrapper (WNE) is implemented as a lightweight normalize-apply-backbone-denormalize module around the chosen backbone.

*Table 4.* **Training hyperparameters.** All runs use on-the-fly patch sampling from the same training corpus and AWGN corruption. For the single-noise diagnostic, $\sigma$ is fixed during training ($\sigma_{\text{train}} \in \{10, 25, 50\}$) and varied at test time. * indicates that the learning rate is halved every 100,000 iterations. Herbreteau et al. (2023) use a constant learning rate for FDnCNN variants, noting that "speed improvements are certainly possible by adapting the learning rate throughout optimization."

| Model / variant | Batch | Patch $P$ | Loss | LR | Iterations |
|---|---|---|---|---|---|
| DnCNN (Baseline) | 128 | 70 | $\ell_1$ | 1e−4 | 500,000 |
| FDnCNN (SE-arch) | 128 | 70 | $\ell_1$ | 1e−4 | 500,000 |
| FDnCNN (NE-arch) | 128 | 70 | MSE | 1e−4 | 900,000 |
| DnCNN (WNE) | 128 | 70 | MSE | 1e−4* | 500,000 |
| SwinIR (Baseline) | 32 | 64 | MSE | 1e−4 | 500,000 |
| SwinIR (WNE) | 32 | 64 | MSE | 1e−4* | 500,000 |

## B. Description of Datasets and Training Details

**Relation to prior work (protocol).** We follow the dataset construction and training protocol of Herbreteau et al. (2023, Appendix B, and implementation notes in Appendix A), which in turn adopts the large training set of Zhang et al. (2022); we use this same training set for all models and experiments. To diagnose noise-level mismatch in a controlled setting, we also consider single-noise training with fixed $\sigma_{\text{train}}$, following the single-noise protocol of Herbreteau et al. (2023).

**Datasets.** Following Herbreteau et al. (2023), our training set is the standard large-scale denoising training corpus popularized by DRUNet/DPIR-style pipelines. The dataset is composed of 8,694 images:

(i) BSD400 (400 images) from the Berkeley Segmentation Dataset (Martin et al., 2001),

(ii) the Waterloo Exploration Database (4,744 images) (Ma et al., 2017),

(iii) DIV2K (900 images) (Agustsson & Timofte, 2017),

(iv) Flickr2K (about 2.6k images) (Lim et al., 2017).

We apply random flips and 90° rotations for data augmentation. We report test results on Set12 and BSD68 (Martin et al., 2001).

**Preprocessing and noise model.** All experiments use additive white Gaussian noise (AWGN). Images are processed in grayscale for Set12/BSD68 evaluation. Pixel intensities are normalized to $[0, 1]$; noise levels $\sigma$ are reported in 8-bit units, so the injected noise standard deviation in $[0, 1]$ units is $\sigma/255$. For the single-noise mismatch diagnostic, we train at a fixed $\sigma_{\text{train}} \in \{10, 25, 50\}$ and evaluate at a range of $\sigma_{\text{test}}$ values.

**Patch sampling.** We train from on-the-fly random crops: each iteration samples a clean training image uniformly, extracts a random $P \times P$ patch, and corrupts it with AWGN at the chosen $\sigma$. Unless stated otherwise, we use $P = 70$ (matching Herbreteau et al. (2023)'s FDnCNN/DnCNN setting).

**Optimization.** We train with Adam and default $\beta$ parameters (Kingma & Ba, 2015). Unless stated otherwise, we use an initial learning rate $10^{-4}$. We train for a fixed number of iterations (not tied to epochs); in our code this corresponds to `num_steps` iterations. For the NE-arch variants we train for longer following Herbreteau et al. (2023). We use MSE for our NE-enforcing variants (NE-arch and WNE) following Herbreteau et al. (2023).

**Model-specific settings.** Table 4 summarizes the training hyperparameters used in our experiments. For DnCNN-family architectural baselines (SE-arch and NE-arch), we follow the corresponding prescriptions from Mohan et al. (2020); Herbreteau et al. (2023). For SwinIR, we use the standard denoising SwinIR backbone (Liang et al., 2021) and train it under the same AWGN patch-based pipeline to keep the comparison controlled.

**Validation.** We report results using the final checkpoint rather than early-stopping on validation PSNR, as our focus is mismatch behavior rather than peak matched-noise performance.

*Table 5.* **Runtime overhead for SwinIR (seconds per batch).** Timing with batch size 32, spatial size $64 \times 64$, warmup $= 3$, and timed steps $= 10$. "Backward" measures one training iteration (forward + loss + backward), and "Inference" measures forward-only. Values in parentheses are ratios relative to the baseline on the same device.

| Variant | GPU | | CPU | |
|---|---|---|---|---|
| | Backward (s) | Inference (s) | Backward (s) | Inference (s) |
| Baseline | 0.271 | 0.101 | 16.31 | 3.52 |
| WNE | 0.271 (1.00×) | 0.101 (1.00×) | 18.22 (1.12×) | 3.81 (1.08×) |

*Table 6.* **Runtime overhead for FDnCNN (seconds per batch).** Timing on inputs of shape $16 \times 1 \times 128 \times 128$ with warmup $= 3$ and timed steps $= 10$. "Backward" measures one training iteration (forward + loss + backward), and "Inference" measures forward-only. Values in parentheses are ratios relative to the baseline on the same device. Hardware: GPU = `NVIDIA GeForce RTX 4090`, CPU = `AMD Ryzen Threadripper PRO 5955WX 16-Cores`.

| Variant | GPU | | CPU | |
|---|---|---|---|---|
| | Backward (s) | Inference (s) | Backward (s) | Inference (s) |
| Baseline | 0.034 | 0.011 | 2.53 | 0.82 |
| WNE | 0.033 (0.99×) | 0.011 (1.00×) | 2.51 (0.99×) | 0.83 (1.01×) |
| NE-arch | 0.054 (1.60×) | 0.019 (1.69×) | 3.79 (1.50×) | 1.57 (1.90×) |

## B.1. Runtime Benchmarking Protocol

We report wall-clock seconds per batch, averaged over 10 timed iterations after 3 warmup iterations, using the same input shapes across variants. "Backward" corresponds to one training iteration (forward + loss + backward) and "Inference" corresponds to forward-only under the `torch.no_grad()` context. Absolute times depend on hardware and system settings, but ratios are informative because all variants share the same backbone family and are timed using the same harness. For SwinIR, we additionally report timings on inputs of shape $32 \times 1 \times 64 \times 64$, reflecting the transformer's training batch and patch size in our setup (Table 5). On CPU, we observe a small but nonzero overhead in our current PyTorch implementation; this may be mitigated with more careful CPU benchmarking. Our focus is GPU training and inference, where modern denoisers are typically deployed.

# C. Mathematical Proofs

## C.1. A Lemma from Herbreteau et al. (2023)

We reproduce the following lemma from Herbreteau et al. (2023) for completeness; it is not used elsewhere and is equivalent to our characterization via Remark 1.

**Lemma 1** (Lemma 1 in Herbreteau et al. (2023)). *Let $f : \mathbb{R}^d \to \mathbb{R}^d$ satisfy normalization equivariance:*

$$f(ay + b\mathbf{1}) = af(y) + b\mathbf{1} \qquad \text{for all } a > 0, \ b \in \mathbb{R}.$$

*Then $f$ is entirely determined by its restriction to*

$$S^{d-1} \cap \text{Span}(\mathbf{1})^{\perp}.$$

*Proof.* First note that NE implies scale equivariance (SE) by taking $b = 0$:

$$f(ay) = af(y) \qquad (a > 0).$$

Fix $x \in \mathbb{R}^d$. Decompose $x$ orthogonally as

$$x = x_1 + x_2, \qquad x_1 \in \text{Span}(\mathbf{1})^{\perp}, \quad x_2 \in \text{Span}(\mathbf{1}).$$

If $x_1 = 0$, then $x = x_2 = \alpha\mathbf{1}$ for some $\alpha \in \mathbb{R}$. Using NE with $y = 0$ gives

$$f(\alpha\mathbf{1}) = f(0 + \alpha\mathbf{1}) = f(0) + \alpha\mathbf{1}.$$

Also, SE forces $f(0) = 0$: $f(0) = f(a \cdot 0) = af(0)$ for all $a > 0$, so $f(0) = 0$. Hence $f(\alpha\mathbf{1}) = \alpha\mathbf{1}$. Thus $f(x)$ is determined on $\mathrm{Span}(\mathbf{1})$.

Assume $x_1 \neq 0$. Using NE with the shift $x_2 \in \mathrm{Span}(\mathbf{1})$,

$$f(x) = f(x_1 + x_2) = f(x_1) + x_2.$$

Now let $z := x_1/\|x_1\|_2$. Then $z \in S^{d-1} \cap \mathrm{Span}(\mathbf{1})^\perp$, and by SE,

$$f(x_1) = f(\|x_1\|_2\, z) = \|x_1\|_2\, f(z).$$

Therefore

$$f(x) = \|x_1\|_2\, f(z) + x_2,$$

so knowing $f$ on $S^{d-1} \cap \mathrm{Span}(\mathbf{1})^\perp$ determines $f(x)$ for every $x \in \mathbb{R}^d$. $\qquad\square$

*Remark* 1 (Connection to our normalized manifold $\mathcal{M}$). Recall

$$\mathcal{M} := \{z \in \mathbb{R}^d : \mu(z) = 0,\ \mathrm{std}(z) = 1\} = \{z \in \mathrm{Span}(\mathbf{1})^\perp : \|z\|_2 = \sqrt{d}\}.$$

Hence

$$\mathcal{M} = \sqrt{d}\big(S^{d-1} \cap \mathrm{Span}(\mathbf{1})^\perp\big).$$

Specifying a function on $\mathcal{M}$ is equivalent to specifying it on $S^{d-1} \cap \mathrm{Span}(\mathbf{1})^\perp$, up to the fixed rescaling by $\sqrt{d}$.

### C.2. Affine Identities for $\mu$, std, and $T_{\mathrm{NE}}$

We include the following proposition for completeness.

**Proposition 1** (Mean and standard deviation under affine transforms). *For all $a > 0$, $b \in \mathbb{R}$, and $y \in \mathbb{R}^d$,*

$$\mu(ay + b\mathbf{1}) = a\,\mu(y) + b, \qquad \mathrm{std}(ay + b\mathbf{1}) = a\,\mathrm{std}(y), \qquad T_{\mathrm{NE}}(ay + b\mathbf{1}) = T_{\mathrm{NE}}(y),$$

*where*

$$T_{\mathrm{NE}}(y) := \begin{cases} \dfrac{y - \mu(y)\mathbf{1}}{\mathrm{std}(y)} & \textit{if } \mathrm{std}(y) > 0, \\ 0 & \textit{if } \mathrm{std}(y) = 0. \end{cases}$$

*Proof.* The mean identity follows from linearity of summation.

For the std, note

$$ay + b\mathbf{1} - \mu(ay + b\mathbf{1})\mathbf{1} = a\big(y - \mu(y)\mathbf{1}\big),$$

so $\mathrm{std}(ay + b\mathbf{1}) = (1/\sqrt{d})\|a(y - \mu(y)\mathbf{1})\|_2 = a\,\mathrm{std}(y)$.

For $T_{\mathrm{NE}}$: If $\mathrm{std}(y) > 0$,

$$T_{\mathrm{NE}}(ay + b\mathbf{1}) = \frac{a(y - \mu(y)\mathbf{1})}{a\,\mathrm{std}(y)} = T_{\mathrm{NE}}(y).$$

If $\mathrm{std}(y) = 0$, then $y$ is constant, hence $ay + b\mathbf{1}$ is constant, and both map to 0 by definition. $\qquad\square$

### C.3. Normalization Equivariance of the Wrapper

**Proposition 2** (NE of the wrapper). *Let $g : \mathbb{R}^d \to \mathbb{R}^d$ be an arbitrary map and define*

$$f^{\mathrm{WNE}}(y) := \mathrm{std}(y)\, g\big(T_{\mathrm{NE}}(y)\big) + \mu(y)\mathbf{1}.$$

*Then $f^{\mathrm{WNE}}$ is normalization equivariant:*

$$f^{\mathrm{WNE}}(ay + b\mathbf{1}) = a\, f^{\mathrm{WNE}}(y) + b\mathbf{1} \qquad \textit{for all } a > 0,\ b \in \mathbb{R}.$$

*Proof.* Using Proposition 1,

$$f^{\text{WNE}}(ay + b\mathbf{1}) = \text{std}(ay + b\mathbf{1}) \, g\big(T_{\text{NE}}(ay + b\mathbf{1})\big) + \mu(ay + b\mathbf{1})\mathbf{1}$$
$$= a \, \text{std}(y) \, g\big(T_{\text{NE}}(y)\big) + (a\mu(y) + b)\mathbf{1}$$
$$= a \, f^{\text{WNE}}(y) + b\mathbf{1}.$$

$\square$

## C.4. Complete Characterization of Normalization Equivariance

**Characterization 1** (Characterization of NE maps). *A function $f : \mathbb{R}^d \to \mathbb{R}^d$ is normalization-equivariant if and only if there exists a function $g : \mathcal{M} \to \mathbb{R}^d$ such that, for all $y$ with $\text{std}(y) > 0$,*

$$f(y) = \text{std}(y) \, g\big(T_{\text{NE}}(y)\big) + \mu(y)\mathbf{1}.$$

*Moreover, for $\text{std}(y) = 0$ one necessarily has $f(y) = \mu(y)\mathbf{1}$, and $g$ is uniquely determined by $f$ on $\mathcal{M}$.*

*Proof.* ($\Leftarrow$) Follows directly from Proposition 2.

($\Rightarrow$) Assume $f$ is NE. Define $g : \mathcal{M} \to \mathbb{R}^d$ as the restriction of $f$ to $\mathcal{M}$, i.e., $g(z) := f(z)$ for $z \in \mathcal{M}$.

If $\text{std}(y) = 0$, then $y = \mu(y)\mathbf{1}$. Scale equivariance (the case $b = 0$ of NE) implies $f(0) = 0$. Applying NE with $a = 1$ to the input $0$ gives

$$f(y) = f(\mu(y)\mathbf{1}) = f(0) + \mu(y)\mathbf{1} = \mu(y)\mathbf{1}.$$

If $\text{std}(y) > 0$, then by definition

$$y = \text{std}(y) \, T_{\text{NE}}(y) + \mu(y)\mathbf{1}, \qquad T_{\text{NE}}(y) \in \mathcal{M}.$$

Applying NE gives:

$$f(y) = f\big(\text{std}(y) \, T_{\text{NE}}(y) + \mu(y)\mathbf{1}\big) = \text{std}(y) \, f\big(T_{\text{NE}}(y)\big) + \mu(y)\mathbf{1} = \text{std}(y) \, g\big(T_{\text{NE}}(y)\big) + \mu(y)\mathbf{1}.$$

Uniqueness: if $g_1, g_2$ both satisfy the factorization, evaluate at $z \in \mathcal{M}$ (where $\mu(z) = 0$, $\text{std}(z) = 1$) to obtain

$$f(z) = g_i(z),$$

so $g_1(z) = g_2(z)$ for all $z \in \mathcal{M}$. $\square$

## C.5. Training Objective Identity

**Proposition 3** (Raw space MSE equals a weighted normalized MSE). *Let $\tilde{y} := T_{\text{NE}}(y)$ and define the matched target transform*

$$T_{\text{NE}}(x \,;\, y) := \begin{cases} \dfrac{x - \mu(y)\mathbf{1}}{\text{std}(y)} & \text{if } \text{std}(y) > 0, \\ 0 & \text{if } \text{std}(y) = 0. \end{cases}$$

*Let $\tilde{x} := T_{\text{NE}}(x; y)$. For the wrapped model $f_\theta^{\text{WNE}}(y) = \text{std}(y) \, g_\theta(\tilde{y}) + \mu(y)\mathbf{1}$, when $\text{std}(y) > 0$,*

$$\big\| f_\theta^{\text{WNE}}(y) - x \big\|_2^2 = \text{std}(y)^2 \, \big\| g_\theta(\tilde{y}) - \tilde{x} \big\|_2^2.$$

*Proof.* Given $\text{std}(y) > 0$, we write $x = \text{std}(y)\tilde{x} + \mu(y)\mathbf{1}$. Then

$$f_\theta^{\text{WNE}}(y) - x = \big(\text{std}(y)g_\theta(\tilde{y}) + \mu(y)\mathbf{1}\big) - \big(\text{std}(y)\tilde{x} + \mu(y)\mathbf{1}\big)$$
$$= \text{std}(y)\big(g_\theta(\tilde{y}) - \tilde{x}\big).$$

Taking squared norms gives the claim. $\square$

## D. Objective-Level Soft NE Versus Exact Parameterization

Because WNE is loss-agnostic, it can in principle be paired with either supervised or self-supervised objectives. Levac et al. (2025) consider a different route: in their noisy-only diffusion-learning setting, NE is encouraged through the training objective rather than imposed by parameterization. In Gaussian denoising, they replace ordinary SURE with an affine-augmented objective. For single-operator inverse problems, they add this denoising term to an equivariant-imaging loss that enforces consistency with the measurement model. The comparison to WNE is therefore between objective-level soft enforcement and exact model-level parameterization.

Porting that objective-level route to a different regime generally requires specifying a new loss, not merely reusing the objective from Levac et al. (2025) verbatim. For example, a supervised analogue is an orbit-averaged regression objective of the form

$$\mathcal{L}_{\text{orbit}} := \mathbb{E}_{(x,y),\alpha,\mu}\big\|f_\theta(\alpha y + \mu\mathbf{1}) - (\alpha x + \mu\mathbf{1})\big\|_2^2,$$

which introduces design choices through the sampling law of the affine variables $(\alpha, \mu)$. Different orbit distributions weight different regions of the affine orbit and therefore define different soft-NE objectives. By contrast, WNE leaves the base loss unchanged and enforces the symmetry exactly by construction around the chosen backbone.

In noisy-only settings, exact parameterization can replace affine augmentation for enforcing NE, but losses such as SURE or equivariant-imaging terms are still needed to provide the self-supervised training signal.

**Our supervised soft-NE implementation.** For the supervised control used here, we instantiate the orbit-averaged objective by sampling a single affine pair $(\alpha, \mu)$ independently for each training image and applying the same transform to both the clean target and its noisy observation:

$$x' = \alpha x + \mu\mathbf{1}, \qquad y' = \alpha y + \mu\mathbf{1}.$$

We then minimize the ordinary supervised regression loss

$$\mathcal{L}_{\text{softNE}} = \mathbb{E}_{(x,y),\alpha,\mu}\big[\|f_\theta(y') - x'\|_2^2\big].$$

In code, this is implemented by redrawing $(\alpha, \mu)$ on each training step, broadcasting each sampled scalar pair over all pixels and channels of the image, and replacing the original pair $(x, y)$ by $(x', y')$ before evaluating the standard MSE loss. Following Levac et al. (2025), we sample $\alpha \sim \mathrm{U}(0, 1)$ and $\mu \sim \mathrm{U}(0, 1)$; unlike a commutator penalty, this introduces no extra loss weight.

**Control experiments.** Figure 8 reports companion PSNR/SSIM curves for two SwinIR controls. The left column compares a multi-noise SwinIR baseline trained on $\sigma \in [0, 55]$, matching the blind DnCNN-B range used by Zhang et al. (2017), against WNE across the same test sweep. The right column compares the supervised soft-NE SwinIR baseline at $\sigma_{\text{train}} = 10$ against WNE under the fixed-$\sigma_{\text{train}}$ mismatch diagnostic. Both controls improve over an unconstrained baseline trained at a single noise level, but neither matches the wrapped model's flat mismatch behavior as cleanly as exact NE. Broader noise coverage or affine augmentation can improve robustness within the sampled range, but they do not imply the algebraic identity $f(ay + b\mathbf{1}) = af(y) + b\mathbf{1}$ for all $a > 0$ and $b \in \mathbb{R}$. The value of WNE is precisely this exact structural guarantee: the affine nuisance is removed analytically rather than approximated from finite orbit coverage. This matters in downstream settings that assume exact NE as part of the algorithmic design, such as the preconditioned PnP analysis of Hong et al. (2024).

## E. Equivariance-Defect Diagnostics for Multi-Noise and Soft-NE Baselines

To complement the PSNR/SSIM mismatch plots, we measure an explicit normalization-equivariance defect

$$\varepsilon_{\text{NE}} = \mathbb{E}_{y,a,b}\left[\frac{\|f(ay + b\mathbf{1}) - (af(y) + b\mathbf{1})\|_2}{\|af(y) + b\mathbf{1}\|_2 + \tau}\right],$$

where the expectation averages over evaluation images and randomly sampled affine perturbations $y \mapsto ay + b\mathbf{1}$, with $a \sim \mathrm{U}(0.5, 1.5)$ and $b \sim \mathrm{U}(-0.25, 0.25)$. Lower is better; exact NE gives zero, and the wrapped implementation is visibly zero at plot resolution in these experiments. Accordingly, $\varepsilon_{\text{NE}}$ asks not whether multi-noise or soft-NE can recover some empirical robustness, but whether they recover the structural identity itself.

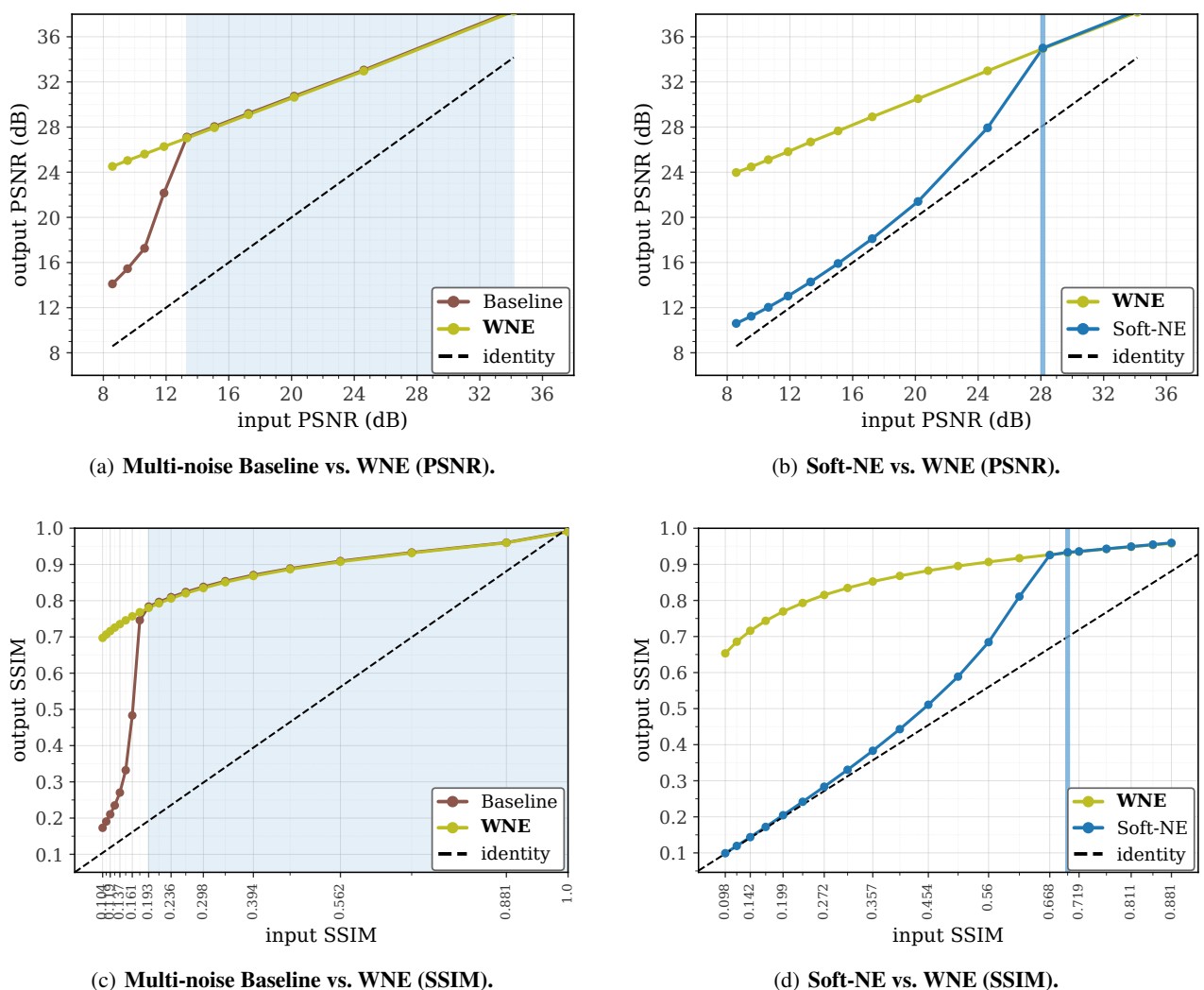

(a) **Multi-noise Baseline vs. WNE (PSNR).**

(b) **Soft-NE vs. WNE (PSNR).**

(c) **Multi-noise Baseline vs. WNE (SSIM).**

(d) **Soft-NE vs. WNE (SSIM).**

*Figure 8.* **PSNR/SSIM controls for multi-noise and supervised soft-NE baselines on SwinIR.** Left column: a SwinIR baseline trained on a range of noise levels $\sigma \in [0, 55]$ against WNE over the same test sweep. Right column: a supervised orbit-averaged soft-NE SwinIR baseline trained at $\sigma_{\text{train}} = 10$ against WNE. Top row uses output PSNR; bottom row uses output SSIM versus input SSIM. In both controls, broader orbit coverage narrows the performance gap, but WNE remains the most stable under mismatch. Figure 9 shows the corresponding explicit equivariance-defect curves.

Figure 9 reports two complementary SwinIR controls. In the left panel, a baseline trained on a range of noise levels $\sigma \in [0, 55]$ has smaller defect inside the training range than outside it, but the defect remains visibly nonzero and increases beyond the range; the wrapped model stays visibly at zero across the sweep. In the right panel, a supervised orbit-averaged soft-NE baseline reduces the defect relative to the unconstrained baseline but does not drive it to zero, whereas WNE again stays visibly at zero.

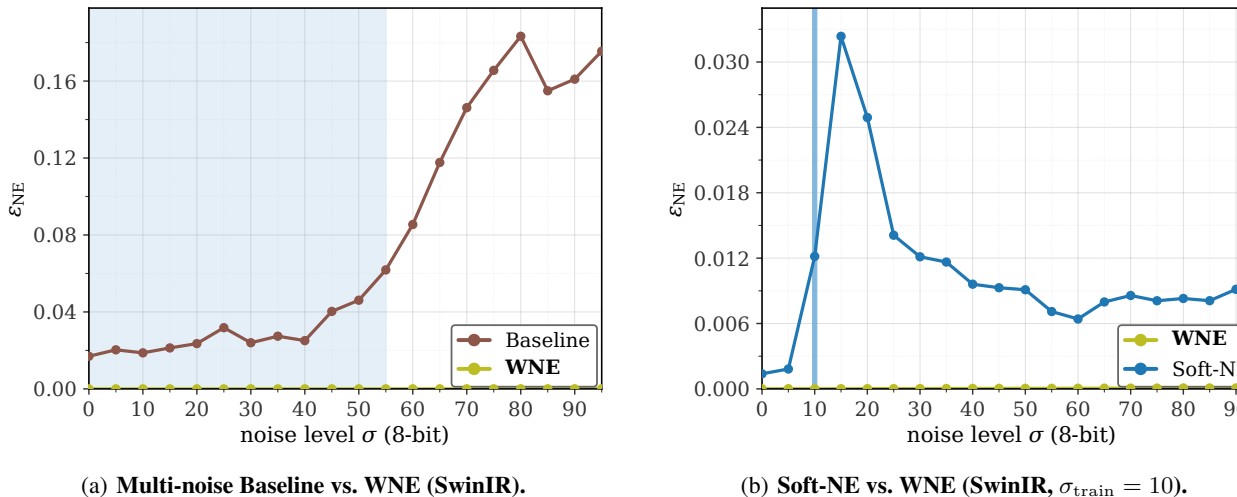

(a) **Multi-noise Baseline vs. WNE (SwinIR).**  (b) **Soft-NE vs. WNE (SwinIR, $\sigma_{\mathrm{train}} = 10$).**

*Figure 9.* **Explicit equivariance-defect diagnostics.** Left: for a multi-noise SwinIR baseline trained on $\sigma \in [0, 55]$ (shaded region), the equivariance defect $\varepsilon_{\mathrm{NE}}$ is lower within the training range but remains visibly nonzero and increases beyond it; WNE stays visibly at zero across the full sweep. Right: for a supervised orbit-averaged soft-NE baseline trained at $\sigma_{\mathrm{train}} = 10$ (vertical line), the defect is reduced but remains visibly nonzero, whereas WNE stays visibly at zero.

## F. Direct Prediction vs Residual Prediction Under the Wrapper

This appendix records two equivalent implementations that yield a normalization-equivariant end-to-end predictor.

**Direct (clean) prediction.** Given $g_\theta$, define

$$\hat{x}_\theta(y) := \mathrm{std}(y)\, g_\theta\big(T_{\mathrm{NE}}(y)\big) + \mu(y)\mathbf{1}.$$

By Proposition 2, $\hat{x}_\theta$ is NE.

**Residual (noise) prediction.** Let $h_\theta$ predict the normalized residual $(y - x)/\mathrm{std}(y)$ in normalized coordinates and define

$$\hat{r}_\theta(y) := \begin{cases} \mathrm{std}(y)\, h_\theta\big(T_{\mathrm{NE}}(y)\big) & \text{if } \mathrm{std}(y) > 0, \\ 0 & \text{if } \mathrm{std}(y) = 0. \end{cases}$$

Output $\hat{x}_\theta(y) := y - \hat{r}_\theta(y)$. Using Proposition 1,

$$\hat{r}_\theta(ay + b\mathbf{1}) = a\, \hat{r}_\theta(y),$$

so

$$\hat{x}_\theta(ay + b\mathbf{1}) = ay + b\mathbf{1} - a\hat{r}_\theta(y) = a\hat{x}_\theta(y) + b\mathbf{1}.$$

Thus $\hat{x}_\theta$ is NE.

**Conversion.** If $f$ is NE, then the residual map $r(y) := y - f(y)$ satisfies: (i) scale equivariance $r(ay) = a\, r(y)$ for all $a > 0$, and (ii) shift-invariance $r(y + b\mathbf{1}) = r(y)$ for all $b \in \mathbb{R}$. Conversely, if a map $r : \mathbb{R}^d \to \mathbb{R}^d$ satisfies (i) and (ii), then $f(y) := y - r(y)$ is NE since for all $a > 0$ and $b \in \mathbb{R}$,

$$f(ay + b\mathbf{1}) = ay + b\mathbf{1} - r(ay + b\mathbf{1}) = ay + b\mathbf{1} - a\, r(y) = af(y) + b\mathbf{1}.$$

This lets you match direct or residual conventions without changing the end-to-end equivariance guarantee.

# G. A Conditional Explanation for Observation 1

We give a two-step explanation for the $\Delta$-driven collapse in Observation 1. Part 1 is an exact per-instance statement. Part 2 lifts it to the population level under one modeling assumption.

**Part 1: Exact per-instance $\sigma$-freeness.** Recall that under AWGN,

$$y = x + \sigma n, \qquad n \sim \mathcal{N}(0, I_d),$$

and that

$$\tilde{y} := T_{\mathrm{NE}}(y), \qquad \tilde{x} := \frac{x - \mu(y)\mathbf{1}}{\mathrm{std}(y)}, \qquad \Delta := \|\tilde{y} - \tilde{x}\|_2.$$

The normalized input $\tilde{y}$ always lies on the normalized manifold $\mathcal{M} := \{z \in \mathbb{R}^d : \mu(z) = 0, \ \mathrm{std}(z) = 1\}$.

**Proposition 4** (Per-instance $\sigma$-freeness). *Fix a nonconstant clean instance $x$. Then conditioning on $\sigma$ is redundant given $(x, \tilde{x}, \Delta)$:*

$$p(\tilde{y} \mid x, \tilde{x}, \Delta = r, \sigma) = p(\tilde{y} \mid x, \tilde{x}, \Delta = r).$$

*Consequently, for any estimator $g$,*

$$\mathbb{E}\big[\|g(\tilde{y}) - \tilde{x}\|_2^2 \mid x, \tilde{x}, \Delta = r, \sigma\big] = \mathbb{E}\big[\|g(\tilde{y}) - \tilde{x}\|_2^2 \mid x, \tilde{x}, \Delta = r\big].$$

*Proof.* Write $a := \mathrm{std}(y)$ and $m := \mu(y)$. Since $\tilde{x} = (x - m\mathbf{1})/a$, fixing $x$ and the realized $\tilde{x}$ determines $a$ and $m$:

$$a = \frac{\mathrm{std}(x)}{\mathrm{std}(\tilde{x})}, \qquad m = \mu(x) - a\,\mu(\tilde{x}).$$

Hence $x = a\tilde{x} + m\mathbf{1}$ and $y = a\tilde{y} + m\mathbf{1}$. Under AWGN,

$$p(y \mid x, \sigma) \propto \exp\left(-\frac{\|y - x\|_2^2}{2\sigma^2}\right).$$

Substituting $y = a\tilde{y} + m\mathbf{1}$ and $x = a\tilde{x} + m\mathbf{1}$ gives

$$p(\tilde{y} \mid x, \tilde{x}, \sigma) \propto \exp\left(-\frac{a^2\|\tilde{y} - \tilde{x}\|_2^2}{2\sigma^2}\right) \mathbf{1}\{\tilde{y} \in \mathcal{M}\},$$

with respect to the natural surface measure on $\mathcal{M}$. Conditioning further on $\Delta = \|\tilde{y} - \tilde{x}\|_2 = r$ makes the exponential factor constant on

$$\mathcal{S}_r(x, \tilde{x}) := \{z \in \mathcal{M} : \|z - \tilde{x}\|_2 = r\}.$$

Therefore the conditional law of $\tilde{y}$ given $(x, \tilde{x}, \Delta = r, \sigma)$ is the normalized surface measure on $\mathcal{S}_r(x, \tilde{x})$, which does not depend on $\sigma$. $\square$

**High-dimensional simplification.** Let

$$\sigma_x := \mathrm{std}(x), \qquad \hat{x} := \frac{x - \mu(x)\mathbf{1}}{\|x - \mu(x)\mathbf{1}\|_2}.$$

In high dimension,

$$\mu(y) \approx \mu(x), \qquad \mathrm{std}(y)^2 \approx \sigma_x^2 + \sigma^2, \qquad \Delta^2 \approx \frac{d\,\sigma^2}{\sigma_x^2 + \sigma^2}.$$

Eliminating $\sigma$ gives $\mathrm{std}(y)^2 \approx d\,\sigma_x^2/(d - \Delta^2)$, and therefore

$$\tilde{x} = \frac{x - \mu(y)\mathbf{1}}{\mathrm{std}(y)} \approx \sqrt{d - \Delta^2}\,\hat{x}.$$

Thus, conditional on $(x, \Delta = r)$, the matched normalized target $\tilde{x}$ is approximately determined, and Proposition 4 yields the approximate per-instance collapse

$$\mathbb{E}\big[\|g(\tilde{y}) - \tilde{x}\|_2^2 \mid x, \Delta = r, \sigma\big] \approx \phi_g(\hat{x}, r).$$

**Part 2: Population-level collapse under one assumption.** To pass from the per-instance statement to the population-level $\Delta$-collapse, we must average over the instance distribution and account for how conditioning on $(\Delta, \sigma)$ changes which clean instances are selected.

**Assumption 1** (Contrast-independence of normalized instance structure). The normalized instance direction

$$\hat{x} := \frac{x - \mu(x)\mathbf{1}}{\|x - \mu(x)\mathbf{1}\|_2}$$

is approximately independent of the empirical contrast $\sigma_x := \mathrm{std}(x)$, in the sense that the conditional law of $\hat{x}$ given $\sigma_x = s$ varies only weakly with $s$ over the range relevant to the analysis.

Assumption 1 is a modeling assumption, not a closed-form consequence of AWGN. It requires that conditioning on empirical contrast does not substantially change the distribution of normalized structure. This is motivated by the image formation process: variation in empirical contrast across natural images reflects both scene content and acquisition conditions such as exposure, gain, and illumination. Standard brightness/contrast augmentations implicitly rely on a similar independence.

**Proposition 5** (Population-level $\Delta$-collapse). *Under Assumption 1, in the high-dimensional regime, for any estimator $g : \mathcal{M} \to \mathbb{R}^d$,*

$$\mathbb{E}\big[\|g(\tilde{y}) - \tilde{x}\|_2^2 \mid \Delta = r, \sigma\big] \approx \psi_g(r),$$

*for some function $\psi_g$ of $r$ alone.*

*Proof.* Apply the tower property:

$$\mathbb{E}\big[\|g(\tilde{y}) - \tilde{x}\|_2^2 \mid \Delta = r, \sigma\big] = \mathbb{E}\Big[\mathbb{E}\big[\|g(\tilde{y}) - \tilde{x}\|_2^2 \mid x, \Delta = r, \sigma\big] \,\Big|\, \Delta = r, \sigma\Big].$$

By the high-dimensional simplification above, the inner conditional expectation is approximately $\phi_g(\hat{x}, r)$. The relation $\Delta^2 \approx d\sigma^2/(\sigma_x^2 + \sigma^2)$ implies that at fixed $(\Delta = r, \sigma)$, conditioning selects a narrow band of contrasts $\sigma_x$. Under Assumption 1, selecting instances through this contrast variable does not substantially change the conditional distribution of $\hat{x}$. Therefore the conditional expectation depends only weakly on $\sigma$ and can be approximated by a function of $r$ alone. $\square$

**Connection to Observation 1.** Proposition 5 should be read as a conditional explanation of Observation 1: the $\Delta$-collapse follows exactly at the per-instance level and lifts approximately to the population level under high-dimensional concentration together with Assumption 1. The residual $\sigma$-spread visible in Figure 6 can arise from finite-dimensional concentration error and residual dependence of normalized structure on contrast.

## H. FDnCNN-Family Comparison

We complement the main-text DnCNN results with an FDnCNN-family comparison in which all four variants share the same FDnCNN base architecture. This brackets one important confound in the main-text comparison: standard DnCNN is used for Baseline and WNE, whereas the architectural SE/NE references are built on FDnCNN. It removes that standard-DnCNN-versus-FDnCNN mismatch while preserving the equivariance-enforcing modifications of the architectural baselines, especially the stronger internal replacements used by NE-arch.

**Protocol.** We follow the same fixed-$\sigma_{\mathrm{train}}$ mismatch diagnostic as in Section 4: for each $\sigma_{\mathrm{train}} \in \{10, 25, 50\}$, models are trained at a single noise level and evaluated on Set12 across varying $\sigma_{\mathrm{test}}$, plotting output PSNR versus input PSNR.

**Results.** Figure 10 and Table 7 show that WNE-FDnCNN is competitive with NE-arch across all three noise levels and that the mismatch curves follow the same qualitative trend as in Figure 3. The comparison does not collapse these methods into the same model class; rather, it shows that after bracketing the standard-DnCNN-versus-FDnCNN mismatch, WNE-FDnCNN still follows a robustness trend close to NE-arch.

## I. Normalized-Space Error Versus Difficulty (Baseline vs WNE)

In this appendix section, we extend the normalized-coordinate analysis of Section 5 by comparing the unmodified Baseline to the wrapped WNE across multiple training noise levels. Our aim is to test two related hypotheses: (i) whether the

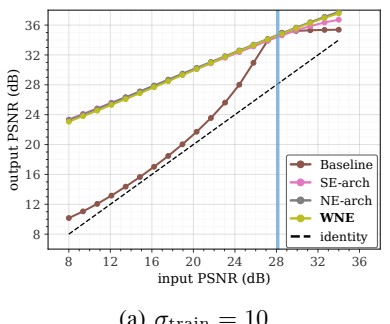
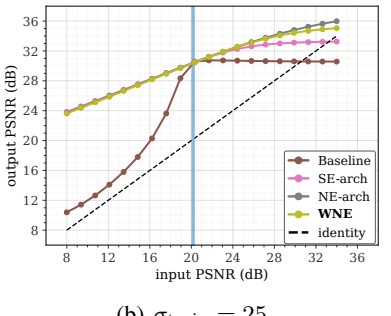
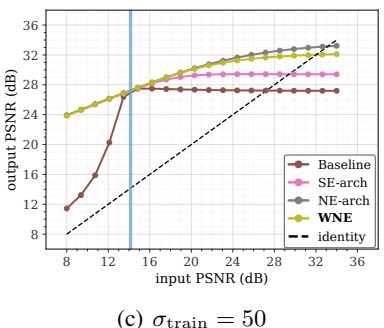

(a) $\sigma_{\text{train}} = 10$        (b) $\sigma_{\text{train}} = 25$        (c) $\sigma_{\text{train}} = 50$

*Figure 10.* **FDnCNN-family comparison under noise-level mismatch.** Output PSNR versus input PSNR on Set12 for models trained at a single $\sigma_{\text{train}}$ and tested at varying $\sigma_{\text{test}}$. All four variants share the same FDnCNN base architecture. WNE closely matches the architectural normalization-equivariant reference NE-arch and improves over SE-arch and the unwrapped baseline. Dashed line: no denoising ($\hat{x} = y$).

*Table 7.* **Matched-noise PSNR (dB) on the FDnCNN backbone.** Average PSNR on Set12 and BSD68 for AWGN with $\sigma \in \{10, 25, 50\}$ (8-bit units). All four variants share the same FDnCNN base architecture. *SE-arch* and *NE-arch* modify internal layers to enforce equivariance, while *WNE* wraps the unmodified backbone.

| | Set12 | | | BSD68 | | |
|---|---|---|---|---|---|---|
| Noise level $\sigma$ (8-bit) | 10 | 25 | 50 | 10 | 25 | 50 |
| *Baseline* | 34.82 | 30.51 | 27.29 | 33.87 | 29.21 | 26.25 |
| *SE-arch* | 34.41 | 30.42 | 27.20 | 33.51 | 29.12 | 26.12 |
| *NE-arch* | 34.69 | 30.42 | 27.26 | 33.77 | 29.13 | 26.21 |
| **WNE** (ours) | 34.62 | 30.38 | 27.23 | 33.72 | 29.11 | 26.19 |

normalized regression error is primarily controlled by the difficulty variable $\Delta = \|\tilde{y} - \tilde{x}\|_2$ rather than by the test noise level $\sigma_{\text{test}}$, and (ii) whether enforcing input-output normalization equivariance strengthens this difficulty-driven behavior. All figures in this section use SwinIR as the backbone.

**Normalized coordinates and difficulty.** Given a clean target $x \in \mathbb{R}^d$ and noisy observation $y = x + \sigma n$, define

$$\tilde{y} := T_{\text{NE}}(y), \qquad\qquad \tilde{x} := T_{\text{NE}}(x\,;\,y) := \frac{x - \mu(y)\mathbf{1}}{\text{std}(y)},$$

$$\delta := \tilde{y} - \tilde{x}, \qquad\qquad \Delta := \|\delta\|_2.$$

We interpret $\Delta$ as a one-dimensional index of denoising difficulty in normalized coordinates (Section 5.2).

**A model-agnostic normalized error.** To compare Baseline and WNE in the same coordinate system, we re-express any model output $\hat{x} = f(y)$ using the instance statistics of the input $y$:

$$\tilde{\hat{x}} := \frac{\hat{x} - \mu(y)\mathbf{1}}{\text{std}(y)}.$$

This does not assume that Baseline is NE; it is a diagnostic that expresses any predictor in the $(\mu(y), \text{std}(y))$-anchored coordinates used by NE. We then measure normalized-space regression quality via

$$Q_f(\tilde{y}, \tilde{x}) := -10 \log_{10} \left\| \tilde{\hat{x}} - \tilde{x} \right\|_2^2,$$

where larger values indicate smaller normalized squared error. For WNE, we have $\tilde{\hat{x}} = g_\theta(\tilde{y})$ by construction, so $Q_f$ coincides with the backbone error in normalized coordinates used in Section 5. For Baseline, $Q_f$ evaluates how well the raw-space prediction aligns with the normalized target $\tilde{x}$ after mapping the output into the $(\mu(y), \text{std}(y))$-anchored coordinates.

**Binning and summary curves.** For each $\sigma_{\text{test}}$, we bin samples by $\Delta$ (fixed-width bins) and plot the empirical mean of $Q_f(\tilde{y}, \tilde{x})$ in each bin, with $\pm 1$ standard deviation shading. All panels share identical axis limits to make cross-$\sigma_{\text{test}}$ comparisons visually consistent.

**Displayed difficulty range (per-curve central mass).**    In each panel, each $\sigma_\text{test}$ curve is displayed only over the central 95% mass of its own test-time $\Delta$ distribution, that is,

$$\Delta \in \big[q_{0.025}(\sigma_\text{test}),\ q_{0.975}(\sigma_\text{test})\big].$$

We do not interpret tail behavior from these plots.

**Results.**    Figure 11 reports six plots arranged by training noise level (rows) and model variant (columns). Across $\sigma_\text{test}$, WNE exhibits an approximate collapse of $Q_f$ versus $\Delta$ within the bulk of each curve, indicating that once difficulty is controlled by $\Delta$, the residual dependence of normalized regression error on $\sigma_\text{test}$ is small. In contrast, Baseline shows a stronger dependence on $\sigma_\text{test}$ at fixed $\Delta$, suggesting that without explicit input-output normalization equivariance, normalized-coordinate behavior is less purely difficulty-driven. Together with the train-test overlap in $\Delta$ (Table 3), these results support the interpretation that improved cross-noise robustness under WNE arises from reduced distribution shift in normalized coordinates and a backbone response that is closer to a difficulty-indexed regression.

## J. SSIM Under Noise-Level Mismatch

We report the Structural Similarity Index (SSIM) (Wang et al., 2004) alongside PSNR for the main DnCNN and SwinIR mismatch experiments. The same fixed-$\sigma_\text{train}$ diagnostic is used as in Section 4, but output quality is measured with SSIM instead of PSNR.

## K. Restormer Backbone Experiments

We use Restormer to test whether the wrapper's robustness benefit extends to a second transformer backbone beyond SwinIR. Restormer uses channel-wise self-attention, a substantially different mechanism from SwinIR's shifted-window attention.

**Protocol.**    We follow the same fixed-$\sigma_\text{train}$ mismatch diagnostic as in Section 4: for each $\sigma_\text{train} \in \{10, 25, 50\}$, models are trained at a single noise level and evaluated on Set12 across varying $\sigma_\text{test}$, plotting output PSNR versus input PSNR. Both variants share the same Restormer backbone described in Appendix A.1, with only the outer WNE wrapper added in the wrapped condition.

**Results.**    Figure 14 shows the same qualitative mismatch pattern as for DnCNN and SwinIR: the unwrapped baseline is noise-level specific, while WNE yields smoother and more stable extrapolation across test noise levels. Table 8 reports matched-noise PSNR; unlike DnCNN and SwinIR, WNE is fully competitive with the baseline at all three noise levels on this backbone.

*Table 8.* **Matched-noise PSNR (dB) on Restormer (grayscale).** Average PSNR on Set12 and BSD68 for AWGN with $\sigma \in \{10, 25, 50\}$ (8-bit units). Both variants share the same Restormer backbone; *WNE* wraps the unmodified architecture.

|  | Set12 | | | BSD68 | | |
| --- | --- | --- | --- | --- | --- | --- |
| Noise level $\sigma$ (8-bit) | 10 | 25 | 50 | 10 | 25 | 50 |
| *Baseline* | 34.79 | 30.76 | 27.65 | 33.71 | 29.29 | 26.42 |
| *WNE* (ours) | 35.05 | 30.79 | 27.69 | 34.00 | 29.38 | 26.45 |

## L. Color Denoising (Restormer on CBSD68)

To test whether the wrapper extends to multi-channel data, we train Restormer with and without WNE on 3-channel color images under the same fixed-$\sigma_\text{train}$ mismatch protocol, evaluating on CBSD68. The wrapper pools $\mu(y)$ and $\text{std}(y)$ jointly over all channels and pixels, matching the global affine action $y \mapsto ay + b\mathbf{1}$ without introducing per-channel statistics.

**Results.**    Figure 15 shows the same qualitative pattern as the grayscale Restormer experiments: the baseline is noise-level specific, while WNE stabilizes performance under mismatch. Table 9 reports matched-noise PSNR on CBSD68; WNE remains competitive with the baseline at all three noise levels.

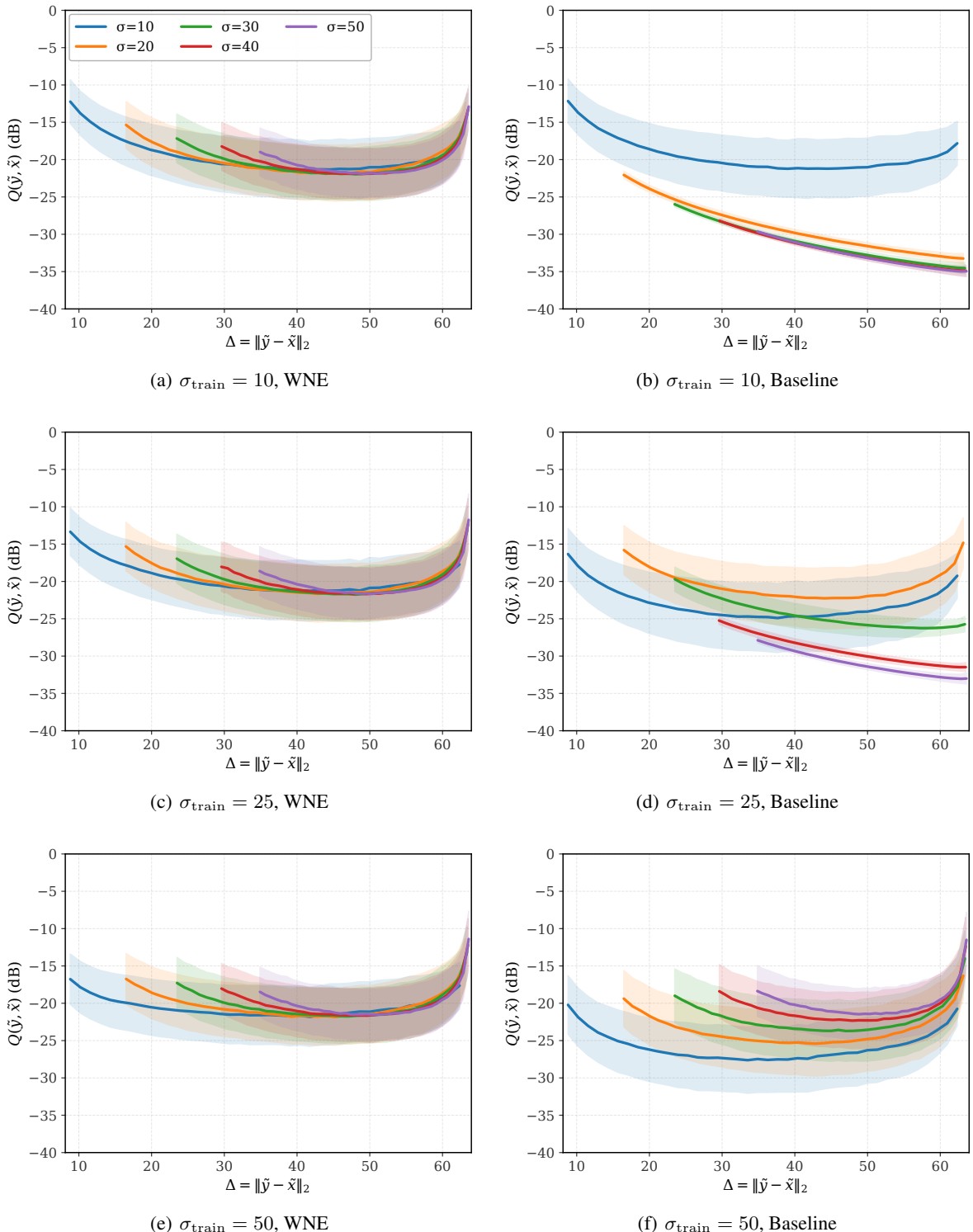

(a) $\sigma_{\text{train}} = 10$, WNE

(b) $\sigma_{\text{train}} = 10$, Baseline

(c) $\sigma_{\text{train}} = 25$, WNE

(d) $\sigma_{\text{train}} = 25$, Baseline

(e) $\sigma_{\text{train}} = 50$, WNE

(f) $\sigma_{\text{train}} = 50$, Baseline

*Figure 11.* **Normalized-space error versus difficulty (Baseline vs WNE).** For each $\sigma_{\text{train}} \in \{10, 25, 50\}$ (rows), we plot the normalized regression quality $Q_f(\tilde{y}, \tilde{x})$ versus difficulty $\Delta = \|\tilde{y} - \tilde{x}\|_2$ for multiple test noise levels $\sigma_{\text{test}}$, comparing WNE (left) to Baseline (right). For readability, each $\sigma_{\text{test}}$ curve is shown only over the central 95% mass of its own test-time $\Delta$ distribution, $\Delta \in [q_{0.025}(\sigma_{\text{test}}), q_{0.975}(\sigma_{\text{test}})]$, and bins outside this range are omitted.

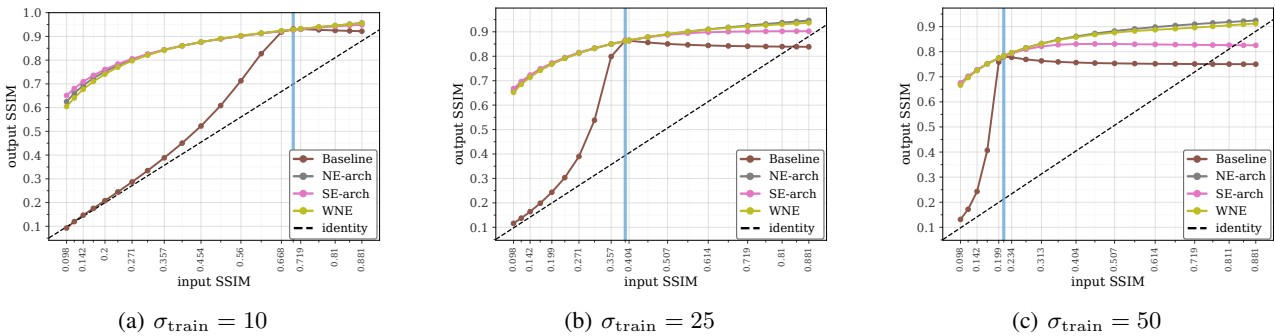

*Figure 12.* **SSIM under noise-level mismatch for DnCNN.** Same mismatch diagnostic as Figure 3, but measured with SSIM instead of PSNR. Across all three training noise levels, WNE preserves the robustness trend of normalization-equivariant models under train-test mismatch, while the Baseline remains noise-level specific.

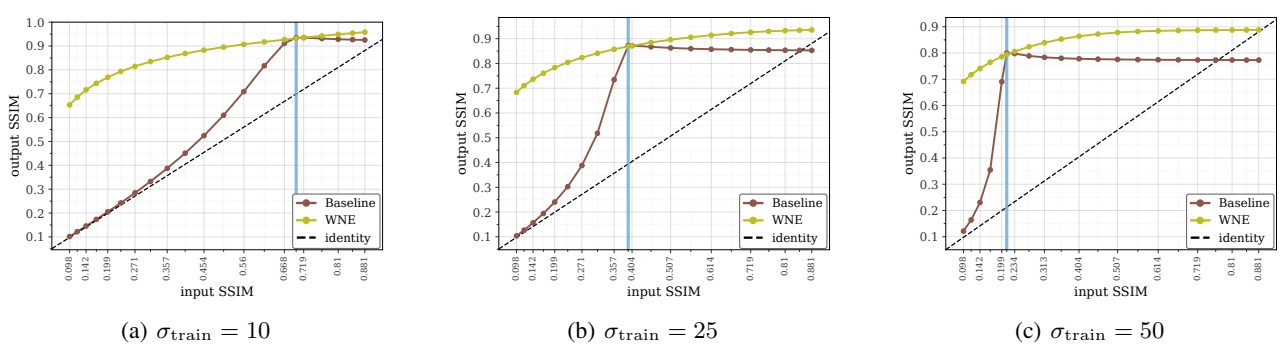

*Figure 13.* **SSIM under noise-level mismatch for SwinIR.** Same mismatch diagnostic as Figure 4, but measured with SSIM instead of PSNR. The Baseline SwinIR degrades once $\sigma_{\text{test}} \neq \sigma_{\text{train}}$, while WNE stabilizes structural quality across a broader range of test noise levels.

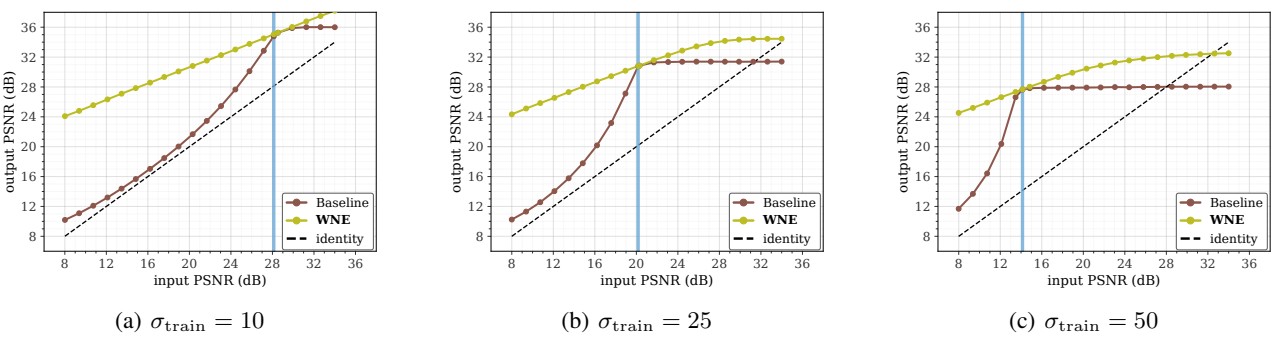

*Figure 14.* **Noise-level mismatch on Restormer.** Same diagnostic as Figures 3–4, applied to a second transformer backbone. The Baseline Restormer degrades under mismatch, while WNE stabilizes performance across test noise levels. Dashed line: no denoising ($\hat{x} = y$).

*Table 9.* **Matched-noise PSNR (dB) on color Restormer (CBSD68).** Average PSNR on CBSD68 for AWGN with $\sigma \in \{10, 25, 50\}$ (8-bit units). Both variants share the same Restormer backbone trained on color patches; *WNE* wraps the unmodified architecture with statistics pooled jointly over all channels and pixels.

|  | CBSD68 | | |
| --- | --- | --- | --- |
| Noise level $\sigma$ (8-bit) | 10 | 25 | 50 |
| *Baseline* | 36.36 | 31.60 | 28.33 |
| *WNE* (ours) | 36.51 | 31.61 | 28.36 |

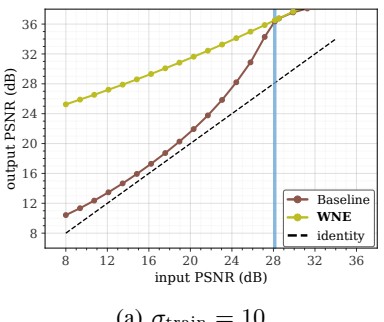 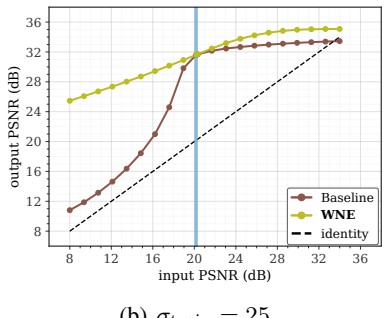 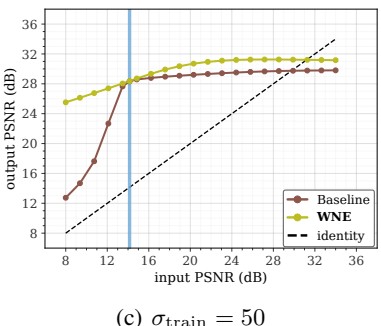

(a) $\sigma_{\text{train}} = 10$       (b) $\sigma_{\text{train}} = 25$       (c) $\sigma_{\text{train}} = 50$

*Figure 15.* **Noise-level mismatch on color Restormer (CBSD68).** Same diagnostic as Figures 3–4, now applied to 3-channel color images. The Baseline Restormer degrades under mismatch, while WNE stabilizes performance across test noise levels. Dashed line: no denoising ($\hat{x} = y$).

## M. Real-Noise Validation on SIDD

We validate that the wrapper does not degrade performance on real sensor noise by training SwinIR Baseline and SwinIR-WNE on the SIDD sRGB track (Abdelhamed et al., 2018). Unlike the controlled fixed-$\sigma_{\text{train}}$ AWGN protocol of Section 4, SIDD exhibits sensor-, ISP-, and scene-dependent noise, so this setting complements rather than replicates the mismatch diagnostic: it tests that enforcing NE remains compatible with a realistic noise distribution rather than isolating the NE mechanism.

**Protocol.** We train SwinIR Baseline and SwinIR-WNE on pre-extracted SIDD sRGB training patches using the same backbone in both conditions, with only the outer wrapper changed. SIDD training uses random $64 \times 64$ crops, batch size 32, AdamW, and a cosine-decayed learning rate from $3 \times 10^{-4}$ to $10^{-6}$ over $400{,}000$ updates. Evaluation uses the standard SIDD validation blocks, corresponding to 1280 sRGB blocks in total. For reporting, predictions and targets are quantized to uint8 sRGB. PSNR is computed with the standard 8-bit formula, and SSIM is computed in RGB with Wang-style settings (Wang et al., 2004).

*Table 10.* **SIDD validation (sRGB).** Average PSNR/SSIM over the 1280 official SIDD validation blocks. Both variants use the same SwinIR backbone; *WNE* only adds the outer normalize-process-denormalize wrapper.

|  | PSNR (dB) | SSIM |
|---|---|---|
| *Baseline* | 39.27 | 0.9150 |
| ***WNE*** (ours) | 39.26 | 0.9156 |

**Results.** Table 10 shows that the two SwinIR variants are matched on SIDD validation: WNE is within 0.01 dB of the baseline in PSNR and essentially identical in SSIM. Combined with the AWGN matched-noise results in Table 1, this indicates that the small matched-AWGN trade-off visible for SwinIR on Set12/BSD68 does not persist on real sensor noise.

## N. Reusing Noise2Noise-Trained Denoisers for Sampling and Inverse Problems

The normalized-coordinate analysis in Section 5 predicts that exact NE should help single-$\sigma$-trained denoisers remain usable across sampler noise schedules: changing raw noise scale primarily shifts which normalized difficulties are queried, rather than changing what the backbone sees at fixed difficulty. We test this prediction by training denoisers with Noise2Noise (N2N) (Lehtinen et al., 2018) at one noise level and reusing the frozen maps for iterative sampling and constrained inverse-problem reconstruction, alongside a one-step mismatch diagnostic.

We train the same SwinIR backbone with the N2N objective: given two independently corrupted observations $y_1 = x + \sigma_{\text{train}}\eta_1$ and $y_2 = x + \sigma_{\text{train}}\eta_2$ of the same image, both variants minimize $\|f_\theta(y_1) - y_2\|_2^2$ at $\sigma_{\text{train}} = 10$. The Baseline-N2N model uses the unwrapped predictor; WNE-N2N wraps the same backbone, changing only the parameterization. This is distinct from N2N-based reconstruction and RED/PnP-style methods that train self-supervised priors for the target inverse problem from split, repeated, or undersampled measurements (Hendriksen et al., 2020; Liu et al., 2020; Desai et al., 2023; Huang et al., 2024); here the learned map is trained only as an ordinary single-step denoiser and then reused unchanged.

Unless otherwise stated, AWGN training and denoising noise levels are reported in 8-bit units; for example, $\sigma = 10$ means $10/255$ when images are scaled to $[0, 1]$. We evaluate the resulting denoisers in three settings: one-step mismatch across unseen test noise levels, residual-stopped iterative denoising, and constrained random inpainting from sparse observations. The unified sampler used for the two iterative settings is given in Appendix N.1.

**One-step mismatch.** Figure 16 applies the same input-output PSNR mismatch diagnostic used in the main experiments to the two N2N-trained models. WNE-N2N is more stable than the unwrapped baseline across the test sweep, including far from $\sigma_{\text{train}}$. The mismatch benefit is therefore not tied to clean-target supervision.

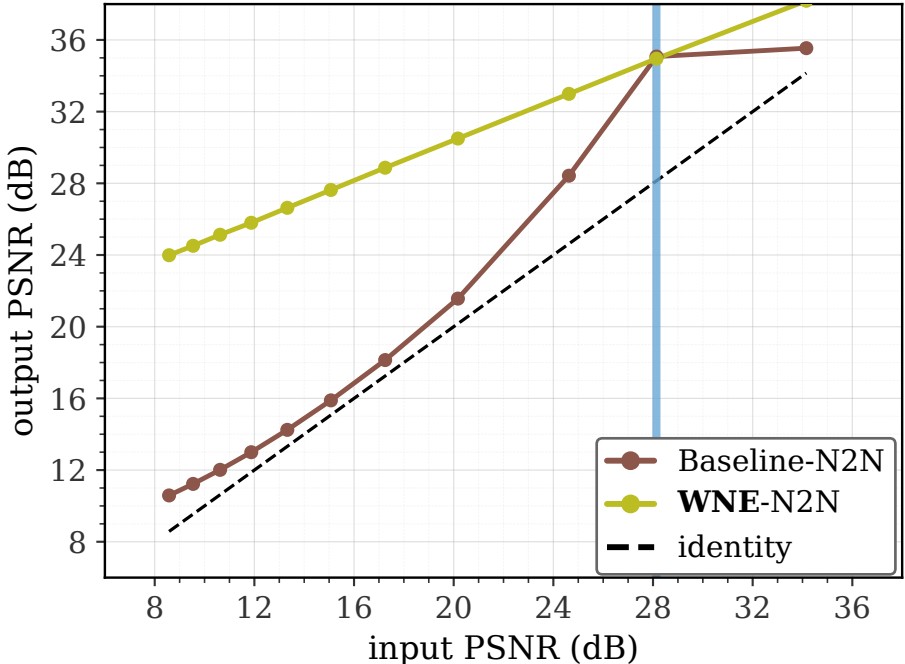

*Figure 16.* **Noise2Noise one-step mismatch check on SwinIR.** Both models are trained with the N2N objective at $\sigma_{\text{train}} = 10$ in 8-bit units and evaluated using the same output-PSNR versus input-PSNR mismatch diagnostic as the main experiments. The wrapped WNE-N2N model is more stable than the unwrapped Baseline-N2N model across the test sweep.

### N.1. Iterative Reuse: A Unified Sampler and the Residual-Stopped Case

Both iterative experiments use the same Kadkhodaie-Simoncelli-style denoiser-residual sampler (Kadkhodaie & Simoncelli, 2021), parameterized by an orthogonal projector $P$ onto an observed measurement subspace. Let $R(y) := D(y) - y$ be the denoiser residual and $x_c$ the observed measurement, with $P = 0$ and $x_c = 0$ for unconstrained denoising and $P = MM^\top$, $x_c = Px^\star$ for measurement-constrained reconstruction. At iterate $y_{t-1}$, the update direction and effective noise level are

$$u_t := (I - P)R(y_{t-1}) + x_c - Py_{t-1}, \qquad \widehat{\sigma}_t := \frac{\|u_t\|_2}{\sqrt{d}}.$$

The sampler updates

$$y_t = y_{t-1} + h_t u_t + \gamma_t z_t, \qquad z_t \sim \mathcal{N}(0, I_d),$$

with step schedule $h_t = h_0 t / (1 + h_0(t - 1))$ and injected-noise amplitude

$$\gamma_t^2 = \left[(1 - \beta h_t)^2 - (1 - h_t)^2\right]\widehat{\sigma}_t^2.$$

For $P = 0$, the direction reduces to $u_t = R(y_{t-1})$ and $\widehat{\sigma}_t = \|R(y_{t-1})\|_2/\sqrt{d}$. This is the update used in the residual-stopped denoising experiment below. Equivalently, it is Algorithm 1 with $P = 0$, using the denoising initialization $y_0 = x^\star + \sigma_{\text{init}}\eta$. For $P = I$, the denoiser term vanishes and the update becomes a pure data-consistency correction; the random-inpainting experiment below uses the intermediate case.

**Why exact NE helps under iterative reuse.** The relevant mechanism is the normalized-coordinate reduction analyzed in Section 5. Under NE, the denoiser factors as

$$D(y) = \text{std}(y)\, g(T_{\text{NE}}(y)) + \mu(y)\mathbf{1}.$$

Changing the raw noise scale therefore does not require learning an unrelated raw-space map; it changes which normalized difficulties $\Delta = \|\tilde{y} - \tilde{x}\|_2$ are queried. Section 5 shows that these difficulty distributions overlap substantially across noise levels and that the normalized error is largely $\Delta$-driven. The N2N sampler experiments stress the same mechanism in an iterative setting: a denoiser trained from noisy pairs at only $\sigma_{\text{train}} = 10$ is repeatedly queried across the sampler's effective noise scales, in the inpainting case starting about $25\times$ above the training scale. WNE does not make the N2N denoiser Bayes-optimal at every scale; it makes the repeated calls use the same normalized regression $g$, with the residual and residual-based stopping scale rescaled analytically. The unwrapped denoiser has no corresponding normalized-coordinate reduction when used off scale.

**Residual-stopped iterative denoising.** We instantiate the unconstrained case ($P = 0$, $\beta = 1$, so $\gamma_t = 0$). Each Set12 image is initialized with AWGN at $\sigma_{\text{init}} = 10$, and the trajectory is stopped when $\widehat{\sigma}_t \leq 1/255$. For each trajectory we record the one-pass denoiser PSNR, the best PSNR attained along the trajectory, and the residual-stopped final PSNR. The one-pass PSNR is computed on $D(y_0)$, whereas trajectory PSNRs are computed on the sampler states $y_t$. The best PSNR is an oracle diagnostic computed using the clean image; it is not used by the sampler. We summarize trajectory collapse by

$$\Delta_{\text{stab}} = \text{PSNR}_{\text{final}} - \text{PSNR}_{\text{best}}.$$

A stable residual stopping rule has $\Delta_{\text{stab}} \approx 0$; a trajectory that peaks early and degrades has a large negative gap. We hold the N2N training regime fixed across both variants: same backbone, same noisy pairs, single noise level. A multi-noise N2N model would require noisy pairs collected or synthesized at multiple noise levels, which is a different data regime. The question is therefore whether exact NE improves off-scale iterative reuse under identical N2N supervision.

*Table 11.* Iterative sampling evaluation on Set12 under N2N training at a single noise level. Both models use the same SwinIR backbone and are trained at $\sigma_{\text{train}} = 10$ in 8-bit units. The sampler is initialized at $\sigma_{\text{init}} = 10$ in 8-bit units and stopped when $\widehat{\sigma}_t \leq 1/255$. Values are means over the 12 Set12 images. The initial noisy PSNR is 28.16 dB for both rows. The best trajectory PSNR is an oracle diagnostic and is not used for stopping.

| Model | One-pass | Best traj. | Final | Final − best | Final − one-pass | Best/final step |
|---|---|---|---|---|---|---|
| Baseline-N2N | 35.07 | 34.20 | 29.49 | −4.71 | −5.58 | 16.67/40.08 |
| WNE-N2N | 34.96 | 34.89 | 34.79 | −0.10 | −0.16 | 22.75/25.00 |

Table 11 shows that the unwrapped baseline is not weaker as a one-step denoiser at the training noise level. Its one-pass PSNR is slightly higher than WNE-N2N (35.07 dB versus 34.96 dB). However, when reused iteratively, the baseline trajectory peaks early and then collapses under the residual stopping rule, ending $4.71$ dB below its best trajectory iterate and $5.58$ dB below its one-pass output. In contrast, WNE-N2N remains stable. The stopped final iterate is only $0.10$ dB below the best trajectory iterate and $0.16$ dB below the one-pass output. The advantage of WNE here is not better one-step PSNR at $\sigma_{\text{train}}$, but predictable global-scale behavior under reuse, consistent with the normalized-coordinate mechanism above.

### N.2. Constrained Random Inpainting

We instantiate the same sampler with a random-inpainting projector $P = MM^\top$ and observation $x_c = Px^\star$. Algorithm 1 gives the full procedure. The reported sampler parameters are on the $[0, 1]$ image scale:

$$\sigma_0 = 1, \qquad \sigma_L = 0.01, \qquad h_0 = 0.01, \qquad \beta = 0.01.$$

Images are scaled to $[0, 1]$, so in 8-bit units this corresponds to $\sigma_0 = 255$ and $\sigma_L = 2.55$. The maximum budget is 1000 iterations.

This setting is deliberately challenging under the fixed N2N training regime. Both denoisers are trained at $\sigma_{\text{train}} = 10$, but the sampler initializes from $\sigma_0 = 1$, about $25\times$ above the training scale. The unwrapped denoiser is therefore queried far outside its training scale at initialization, while WNE normalizes the input and exactly rescales the output residual.

We evaluate random inpainting on Set12 with only $10\%$ observed pixels. Final reconstructions are projected onto the measurement affine set before computing PSNR and SSIM, so both methods satisfy the measurements exactly at evaluation

---

**Algorithm 1** Kadkhodaie-Simoncelli linear inverse sampler

---

**Require:** Denoiser $D$, projector $P$, observation $x_c = Px^\star$, parameters $\sigma_0, \sigma_L, h_0, \beta, T_{\max}$
**Ensure:** Projected reconstruction $\hat{x}$
 1: Draw $z \sim \mathcal{N}(0, I_d)$
 2: $y \leftarrow x_c + 0.5(I - P)\mathbf{1} + \sigma_0 z$
 3: **for** $t = 1, \ldots, T_{\max}$ **do**
 4: $\quad u \leftarrow (I - P)\big(D(y) - y\big) + x_c - Py$
 5: $\quad \widehat{\sigma} \leftarrow \|u\|_2 / \sqrt{d}$
 6: $\quad$ **if** $\widehat{\sigma} \le \sigma_L$ **then**
 7: $\quad\quad$ **break**
 8: $\quad$ **end if**
 9: $\quad h \leftarrow h_0 t / (1 + h_0(t - 1))$
10: $\quad \gamma \leftarrow \Big((1 - \beta h)^2 - (1 - h)^2\Big)^{1/2} \widehat{\sigma}$
11: $\quad$ Draw $z \sim \mathcal{N}(0, I_d)$
12: $\quad y \leftarrow y + hu + \gamma z$
13: **end for**
14: **return** $\hat{x} = x_c + (I - P)y$

---

time. The one-pass column applies the denoiser once to the sampler initialization $y_0$ and evaluates the projected estimate $x_c + (I - P)D(y_0)$.

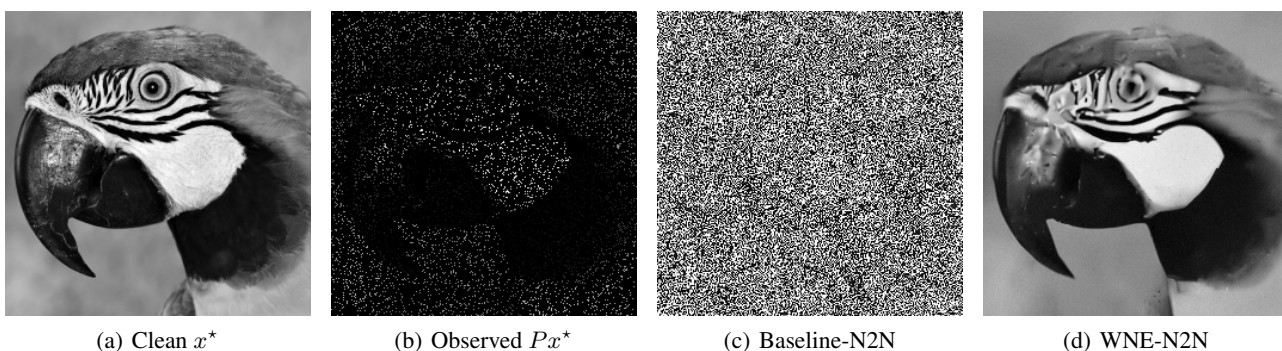

| (a) Clean $x^\star$ | (b) Observed $Px^\star$ | (c) Baseline-N2N | (d) WNE-N2N |

*Figure 17.* **Reusing Noise2Noise-trained denoisers in a constrained inverse-problem sampler.** Both SwinIR denoisers are trained from noisy pairs at a single noise level, $\sigma_{\text{train}} = 10$, then reused in a random-inpainting sampler with 10% observed pixels. The Baseline-N2N denoiser stays close to the sparse observation, while WNE-N2N gives a coherent reconstruction.

*Table 12.* Kadkhodaie-Simoncelli constrained sampler on Set12 random inpainting with 10% observed pixels. Both denoisers are trained with N2N at a single noise level, $\sigma_{\text{train}} = 10$ in 8-bit units. The sampler uses $\sigma_0 = 1$, $\sigma_L = 0.01$, $h_0 = 0.01$, and $\beta = 0.01$. WNE-N2N reaches 23.87 dB versus 6.08 dB for the unwrapped baseline. Numbers are means over Set12, with standard deviations in parentheses. The column "gap" reports final PSNR minus the best PSNR along the trajectory.

| Method | One-pass | Final | SSIM | Gap | Steps |
|---|---|---|---|---|---|
| Observed $Px^\star$ | — | 5.92 (0.93) | 0.020 (0.008) | — | — |
| Baseline-N2N | 6.87 (0.28) | 6.08 (0.27) | 0.009 (0.002) | −0.91 (0.03) | 96.8 (1.3) |
| WNE-N2N | 14.21 (1.31) | **23.87** (2.62) | **0.716** (0.056) | **0.00** (0.00) | 277.6 (26.1) |

Table 12 shows that, under this high-noise initialization and fixed sampler configuration, the unwrapped denoiser stays near the observed projection and ends at 6.08 dB. WNE makes the same denoiser usable inside the sampler and reaches 23.87 dB, an improvement of about 17.8 dB using the same backbone, training pairs, and sampler hyperparameters. The result instantiates the normalized-coordinate mechanism in the inverse-problem regime. The original sampler is a coarse-to-fine procedure initialized at full-scale Gaussian noise: natural for the sampler, but far outside the N2N training scale. Under

this initialization, the vanilla denoiser produces an uninformative residual. WNE preserves the same training objective and architecture, but routes every call through the same normalized regression $g$, so the denoiser can be reused from the high-noise initialization and called across the sampler's noise schedule without changing the sampler hyperparameters.

## O. Input-Only Normalization Versus the Wrapper (IN vs WNE)

In this appendix section we test whether applying an instance-normalization style transform at the input only can replicate the robustness benefits of the full input-output NE wrapper. All figures in this section use a DnCNN backbone and single-noise training at $\sigma_{\text{train}} = 10$. Recall the NE normalization map $T_{\text{NE}}$ from Section 3 and let $g_\theta$ be a standard DnCNN backbone.

**WNE.** We use the wrapped predictor

$$f_\theta^{\text{WNE}}(y) := \text{std}(y)\, g_\theta(T_{\text{NE}}(y)) + \mu(y)\mathbf{1},$$

which is normalization-equivariant by construction (Proposition 2).

**IN (input-only).** We define the input-only normalization baseline by applying the same normalization at the input and not inverting it:

$$f_\theta^{IN}(y) := g_\theta(T_{\text{NE}}(y)).$$

The map $T_{\text{NE}}(y) = (y - \mu(y)\mathbf{1})/\text{std}(y)$ has the same algebraic form as instance normalization (Ulyanov et al., 2016). Our statistics $\mu(y)$ and $\text{std}(y)$ are pooled jointly over all entries (all channels and pixels), rather than per-channel, because the NE group action $y \mapsto ay + b\mathbf{1}$ acts identically on all entries.

**Structural limitation of IN.** Because $T_{\text{NE}}$ is invariant to global contrast and brightness,

$$T_{\text{NE}}(ay + b\mathbf{1}) = T_{\text{NE}}(y) \qquad (a > 0,\ b \in \mathbb{R})$$

(Proposition 1), the input-only baseline satisfies

$$f_\theta^{IN}(ay + b\mathbf{1}) = f_\theta^{IN}(y),$$

so it discards global shift and scale information that an NE denoiser must propagate to the output. In contrast, WNE re-injects $\mu(y)$ and $\text{std}(y)$ analytically at the output.

**Protocol.** We follow the same noise-level mismatch diagnostic as in Section 4: for each $\sigma_{\text{test}}$, we plot output PSNR versus input PSNR on Set12.

**Results.** Figure 18 shows that IN has lower PSNR than Baseline, including at matched noise ($\sigma_{\text{test}} = \sigma_{\text{train}}$), while WNE remains stable under mismatch. Figure 19 illustrates the same effect qualitatively.

## P. Non-Gaussian Corruptions

This appendix tests whether the robustness benefits of enforcing NE extend beyond AWGN. We consider three additive non-Gaussian noises (Uniform, Laplace, Rayleigh) and a non-additive corruption (JPEG artifacts), using the same input-output PSNR diagnostic as in Section 4. We evaluate both the DnCNN family and SwinIR. For the DnCNN-family architectural references, FDnCNN is used for SE-arch and NE-arch, while the standard DnCNN backbone is kept for Baseline and WNE. For SwinIR, we compare Baseline versus WNE.

**Noise models.** For the additive cases we generate $y = x + \sigma\varepsilon$, where $\varepsilon$ is scaled to have unit standard deviation and $\sigma$ controls corruption strength. Uniform and Laplace noises are zero-mean. For Rayleigh, we do not recenter the noise: it has nonzero mean even after scaling to unit standard deviation. Thus this Rayleigh setting introduces a systematic positive brightness bias in expectation, in addition to pixelwise randomness. JPEG artifacts are produced by compressing and decompressing the image at a chosen quality factor.

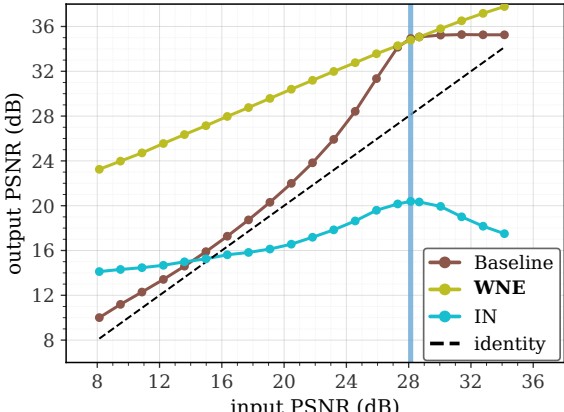

*Figure 18.* **DnCNN: input-output PSNR diagnostic at $\sigma_{\text{train}} = 10$ (IN vs WNE).** Output PSNR versus input PSNR on Set12 as $\sigma_{\text{test}}$ varies. The vertical line marks $\sigma_{\text{train}} = 10$. The dashed identity line corresponds to no denoising ($\hat{x} = y$). Input-only normalization (IN) performs poorly even at matched noise and degrades further under mismatch, while WNE remains robust.

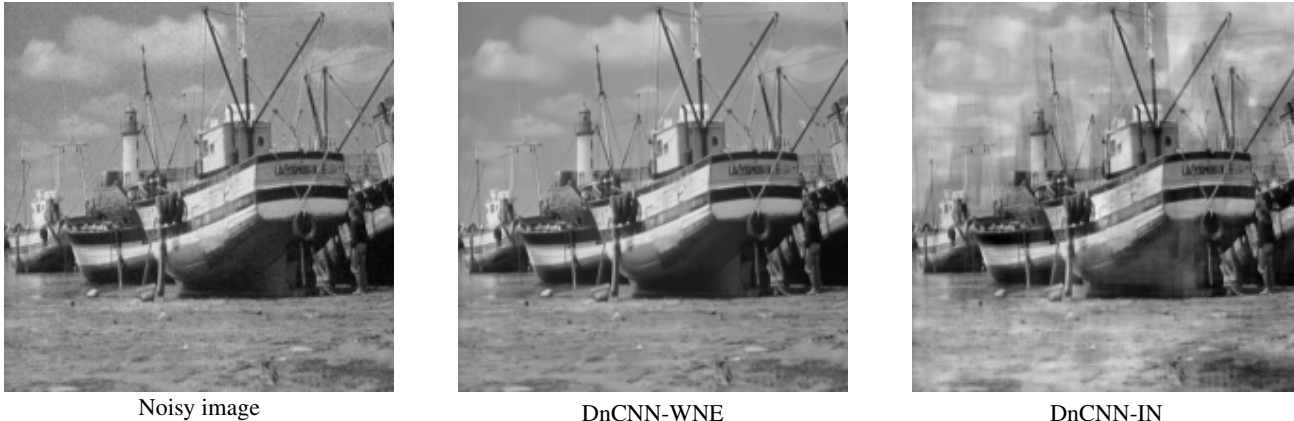

<table>
<tr><td>Noisy image
$\sigma = 10$</td><td>DnCNN-WNE</td><td>DnCNN-IN</td></tr>
</table>

*Figure 19.* **DnCNN qualitative comparison (train $\sigma_{\text{train}} = 10$).** The IN baseline applies $T_{\text{NE}}$ only at the input (no analytic inverse), discarding global brightness and contrast information, and fails severely. The WNE wrapper re-injects $\mu(y)$ and $\text{std}(y)$ at the output and remains qualitatively stable.

**Protocol.** For the additive noises, each model is trained at a single corruption strength (vertical reference line) and evaluated at multiple strengths by varying $\sigma_{\text{test}}$. We plot output PSNR versus input PSNR on Set12; the dashed identity line corresponds to no denoising ($\hat{x} = y$). For JPEG, we plot output PSNR versus quality factor, and we evaluate a simple transfer setting by applying the model trained on Uniform noise to JPEG-corrupted inputs.

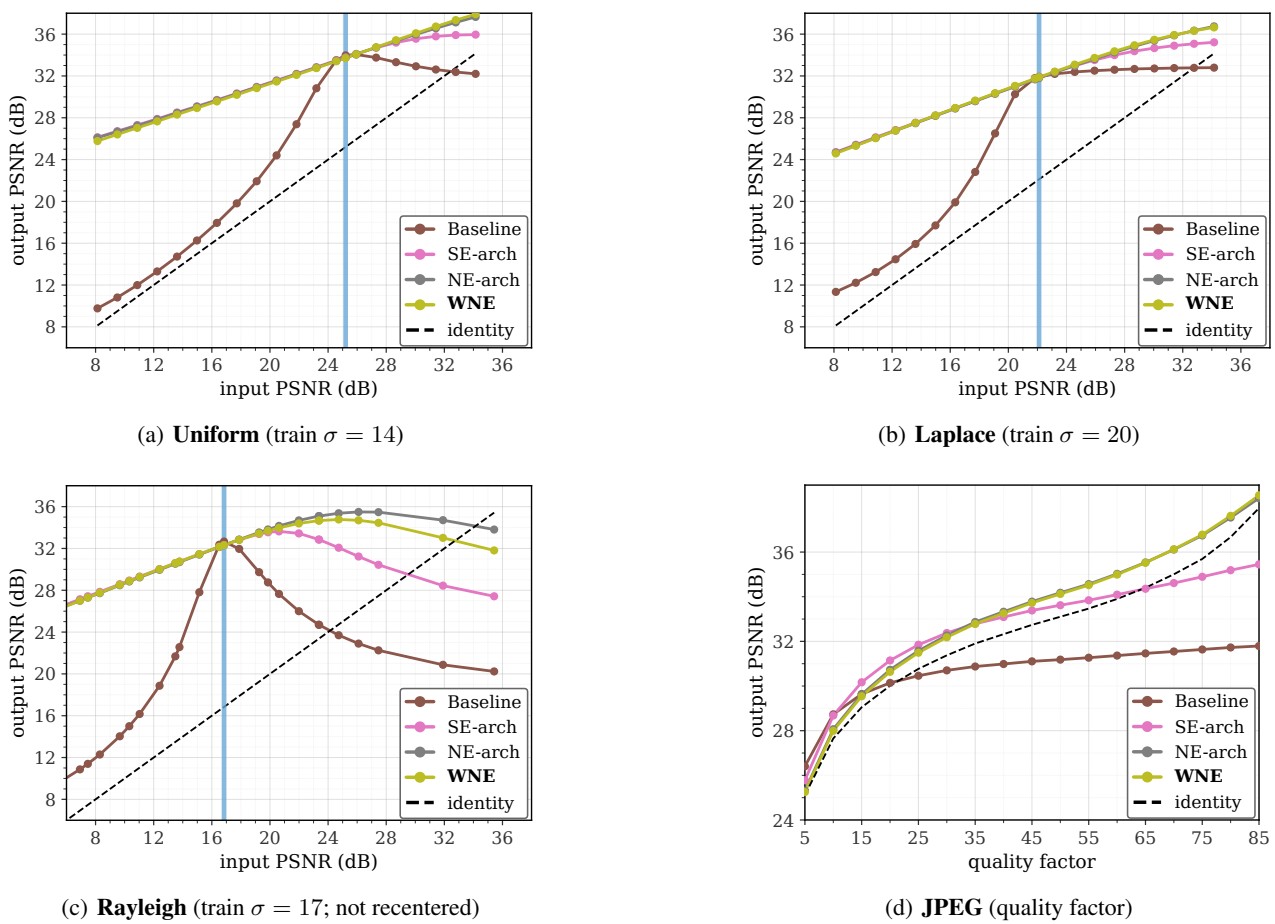

(a) **Uniform** (train $\sigma = 14$)

(b) **Laplace** (train $\sigma = 20$)

(c) **Rayleigh** (train $\sigma = 17$; not recentered)

(d) **JPEG** (quality factor)

*Figure 20.* **Non-Gaussian corruptions (DnCNN family).** Input-output PSNR diagnostic on Set12 for three additive non-Gaussian noises (Uniform, Laplace, Rayleigh) and robustness to JPEG artifacts. All models are blind. For additive noises, the vertical reference line marks the training corruption strength and moving away corresponds to testing at unseen strengths. The dashed identity line is the no-denoising baseline $\hat{x} = y$. Rayleigh noise is not recentered in our implementation (nonzero mean), so it also induces an expected brightness bias. For JPEG, we plot output PSNR versus quality factor and evaluate transfer by applying the model trained on Uniform noise to JPEG-corrupted inputs.

**Results on the DnCNN family.** Figure 20 summarizes the DnCNN-family findings. For Uniform and Laplace, WNE closely matches the architectural normalization-equivariant baseline NE-arch and improves over SE-arch and the unwrapped baseline. For JPEG artifacts, WNE also remains competitive and yields a consistent improvement over the baseline across quality factors in this setting. Rayleigh is the only setup where WNE results in a lower PSNR curve than NE-arch. This is also the only noise model here that is not zero-mean in our implementation (we do not recenter the Rayleigh samples).

**Results on SwinIR.** Figure 21 shows the same extension on the transformer backbone. Across Uniform, Laplace, Rayleigh, and JPEG corruptions, WNE improves over the SwinIR baseline, consistent with the DnCNN-family results above.

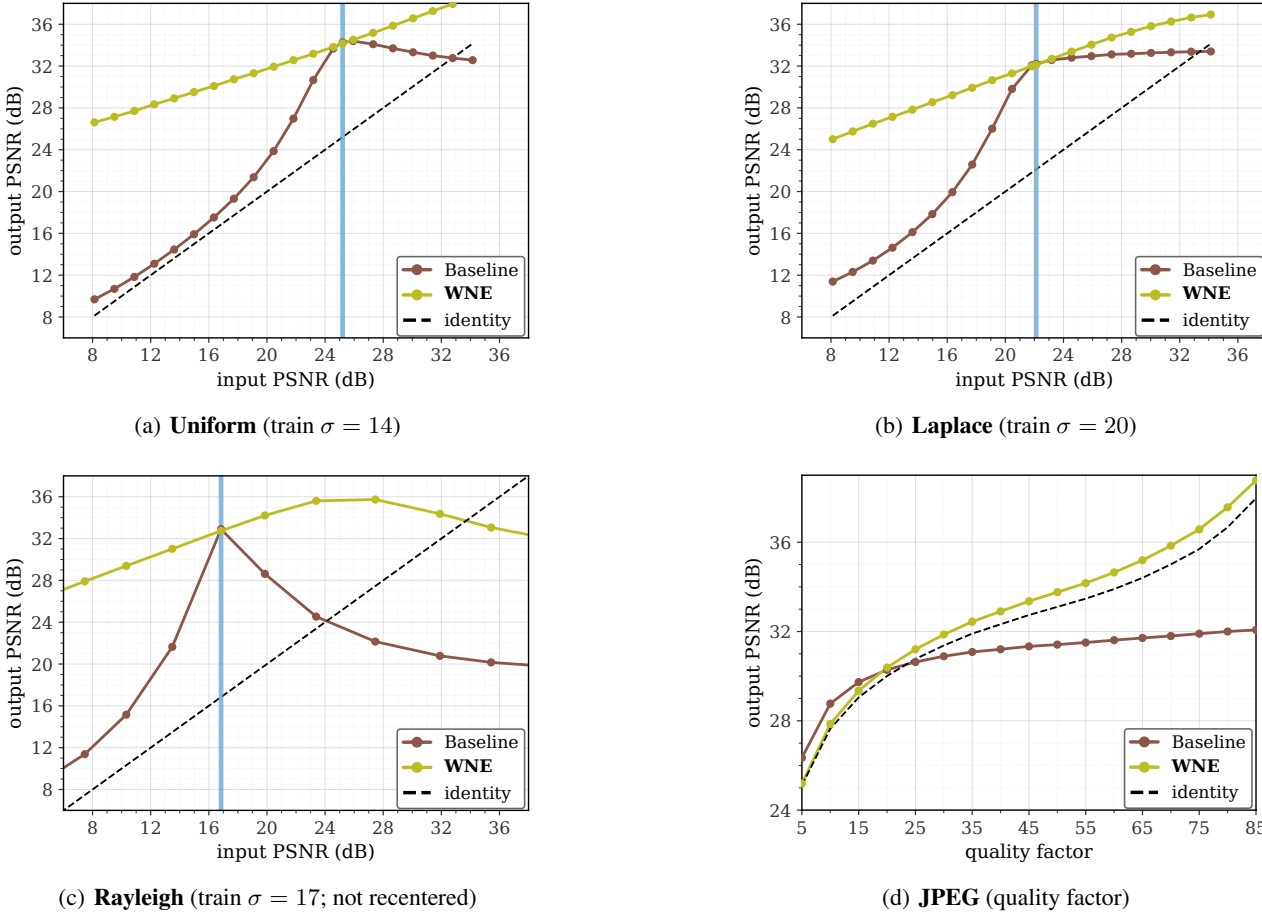

(a) **Uniform** (train $\sigma = 14$)

(b) **Laplace** (train $\sigma = 20$)

(c) **Rayleigh** (train $\sigma = 17$; not recentered)

(d) **JPEG** (quality factor)

*Figure 21.* **Non-Gaussian corruptions (SwinIR).** Same diagnostic as Figure 20, applied to SwinIR Baseline versus SwinIR-WNE. WNE improves over the baseline across all four corruption types, consistent with the DnCNN-family results.

