# OpenReview forum: "Normalization Equivariance for Arbitrary Backbones, with Application to Image Denoising"
_ICML.cc/2026/Conference — ICML 2026 regular_

### Official Review · Reviewer_t8dJ · 2026-03-06

**Soundness:** 3
**Presentation:** 3
**Significance:** 2
**Originality:** 2
**Overall Recommendation:** 3
**Confidence:** 3

**Summary:**

The paper proposes a parameter-free wrapper called WNE that enforces input-output normalization equivariance around any backbone architecture for image denoising tasks. The key insight is that a function is normalization-equivariant if and only if it admits a normalize-process-denormalize factorization, allowing the wrapper to be applied to any network without modifying its internal structure.
Experiments on DnCNN and SwinIR demonstrate that the WNE wrapper improves robustness to noise-level mismatch while maintaining competitive performance at matched noise levels, with negligible computational overhead of approximately 1.00x for both forward and backward passes compared to architectural NE baselines.
The paper introduces a difficulty measure Δ = ||ỹ - x̃||₂ and demonstrates that normalized regression error is largely driven by this difficulty index rather than the absolute noise level, providing insight into the mechanism underlying NE robustness and the asymmetric robustness behavior observed in experiments.

**Compliance With Llm Reviewing Policy:**

Affirmed.

**Key Questions For Authors:**

(1) How does the wrapper perform on images with non-stationary noise or spatially varying statistics? The paper does not adequately analyze scenarios where the WNE wrapper might fail or underperform, particularly for images with extreme contrast or brightness variations that may affect the normalization statistics.
(2) What is the magnitude of NE violation introduced by the numerical stabilization approximation using std_ε(y) = std(y) + ε? The paper does not analyze how this approximation affects the theoretical NE guarantee or under what conditions the approximation significantly deviates from the ideal wrapper.
(3) Can the method be extended to other image restoration tasks beyond denoising such as super-resolution, deblurring, or inpainting? The lack of evaluation beyond denoising raises questions about the broader applicability of the proposed normalization equivariance framework.
(4) Why does the normalized regression error collapse to a Δ-driven function? Observation 1 is stated as an empirical finding without theoretical justification, and the paper would benefit from providing theoretical analysis explaining this collapse phenomenon.

**Limitations:**

(1) The evaluation relies exclusively on PSNR as the quality metric and uses only Set12 and BSD68 as test datasets. Modern image processing research typically includes additional metrics (SSIM, LPIPS) and larger, more diverse test datasets to validate the generalizability of findings.
(2) The paper does not provide clear guidance on when practitioners should use the WNE wrapper versus architectural NE approaches or other robustness-enhancing methods. What are the trade-offs and when is the wrapper preferred over training with diverse noise levels?
(3) The paper lacks comparison with several relevant baselines including data augmentation strategies for improving robustness to noise-level mismatch, meta-learning or domain adaptation approaches for handling distribution shift, and the objective-level soft NE regularization mentioned in related work.
(4) The code availability statement is missing. Providing code would enhance reproducibility and adoption of the proposed method.

**Strengths And Weaknesses:**

Strength:
1. The paper provides a rigorous mathematical characterization of normalization equivariance with sound theoretical proof. The proof that NE maps admit a unique normalize-process-denormalize factorization is mathematically rigorous and provides valuable insight into the structure of equivariant functions.
2.The proposed WNE wrapper is parameter-free and can be applied to any backbone architecture without modification, including modern transformers with attention mechanisms and LayerNorm. This is a significant advantage over prior architectural approaches that require specialized layer designs.
3.The experimental results demonstrate negligible computational overhead on GPU with approximately 1.00x for both forward and backward passes, while architectural NE baselines show up to 1.6x slowdown, making the approach attractive for practical deployment.
Weakness：
1.The experimental evaluation is narrowly focused on the blind denoising task under AWGN noise-level mismatch. While the authors mention applications to inverse problems and diffusion pipelines, no experimental validation is provided for these claimed application domains.
2.The core technical contribution—the WNE wrapper—is essentially a straightforward application of the normalize-process-denormalize template that has been widely used in other domains such as RevIN for time-series forecasting, adding little novelty beyond applying this well-known template.

---

> ### Author Rebuttal · Authors · 2026-03-30
>
> We thank Reviewer t8dJ for the thorough review.
>
> **W1: Narrow experimental focus.**
>
> We have broadened the experiments:
> (1) Restormer (transposed attention) with and without WNE;
> (2) SwinIR non-Gaussian evaluation (Uniform, Laplace, Rayleigh, JPEG);
> (3) color denoising with Restormer on CBSD68.
> All new results are in the anonymous link.
> The revised submission covers two CNN backbones (DnCNN, FDnCNN), two transformer backbones (SwinIR, Restormer), four noise types beyond AWGN, and both greyscale and color.
>
> **W2: WNE is a straightforward application of RevIN.**
>
> The normalize–process–denormalize template appears in RevIN as an engineering heuristic without theoretical justification or connection to equivariance.
> Our contributions go beyond applying this template:
>
> *Characterization.*
> Characterization 1 proves that a map is NE *if and only if* it admits this factorization.
> Input–output wrapping is not an approximation of architectural NE but an equivalent reparameterization of the full NE function class.
> This is absent from RevIN.
>
> *Simplification.*
> Herbreteau et al. (2023) enforce NE by constraining every internal layer, incurring $1.6\times$ runtime cost and ruling out attention and LayerNorm.
> The iff characterization shows that this internal machinery is not required to realize the full NE function class: input–output wrapping achieves the same guarantee with zero overhead on any backbone.
>
> *Analysis.*
> Section 5 provides a mechanism explaining *why* NE improves robustness via the difficulty variable $\Delta$.
> We have also added a new conditional explanation for Observation 1, clarifying when and why the normalized-space MSE becomes primarily $\Delta$-driven (anonymous link, Section R1).
>
> **Q1: Extreme contrast / spatially varying statistics.**
>
> The wrapper is designed to handle *global* affine changes exactly: for any $a>0$ and $b\in\mathbb{R}$, it satisfies $f(ay + b\mathbf{1}) = af(y) + b\mathbf{1}$.
> Equivalently, if two inputs differ only by a global contrast or brightness change, the wrapped backbone receives the same normalized content.
> Thus, for the same underlying image content, extreme global contrast or brightness does not create a different denoising problem for $g$; it only changes the analytic re-scaling applied outside the backbone.
>
> Beyond this exact guarantee, Appendix G shows that WNE remains robust, and substantially improves over the Baseline, even for JPEG artifacts, which are spatially non-stationary.
>
> **Q2: Epsilon stabilization.**
>
> The $\epsilon$-stabilization only affects the denominator in the normalization step, so the deviation from exact NE is negligible whenever $\mathrm{std}(y) \gg \epsilon$.
> With $\epsilon=10^{-5}$, this is the case for natural images in our experiments, so the implementation remains NE to numerical precision.
>
> **Q3: Other restoration tasks.**
>
> The NE prior is relevant whenever the degradation commutes with global affine transforms.
> This is why SE/NE denoisers have already been used beyond denoising in later work, including inverse problems and diffusion-style pipelines.
> We leave direct validation of these settings with WNE to future work.
>
> **Q4: Theoretical justification for Observation 1.**
>
> The revised manuscript adds a conditional explanation for Observation 1 (anonymous link, Section R1).
> At the per-instance level, the residual dependence on $\sigma$ disappears exactly after conditioning on $(x,\tilde{x},\Delta)$.
> At the population level, a high-dimensional approximation together with a contrast-independence assumption yields an approximate $\Delta$-driven collapse of normalized-space MSE, with only weak residual dependence on $\sigma$.
>
> **L1: Metrics and datasets.**
>
> We now report SSIM for all main experiments (anonymous link, Figures R2, R3) and add color benchmarks (Restormer on CBSD68).
> The new results confirm that the robustness gains under mismatch are not accompanied by structural degradation.
>
> **L2: Practical guidance.**
>
> WNE is a natural default when one wants exact NE on an unmodified backbone with negligible overhead.
> Architectural NE is preferred only when constrained internals are desired for other reasons (e.g., Lipschitz control), accepting $1.6\times$ overhead.
> WNE and multi-noise training are orthogonal and can be combined.
>
> **L3: Missing baselines.**
>
> Multi-noise training is orthogonal to WNE.
> Our single-noise diagnostic isolates the structural effect of the NE prior from training-distribution coverage, following Mohan et al. (2020) and Herbreteau et al. (2023).
> Meta-learning and domain adaptation operate at a different level (training strategy vs. structural constraint) and are not directly comparable.
>
> **L4: Code availability.**
>
> Code is included in the supplementary material on OpenReview and will be released publicly upon acceptance.
>
> We would be grateful if the reviewer could consider revising their assessment in light of these changes.
>
> Anonymous link: https://anonymous.4open.science/r/rebuttal_icml2026-04FC/rebuttal.pdf

---

> > ### Author Rebuttal · Reviewer_t8dJ · 2026-04-03
> >
> > The authors do not provide a reasonable justification for this issue: the paper lacks comparison with several relevant baselines including data augmentation strategies for improving robustness to noise-level mismatch, meta-learning or domain adaptation approaches for handling distribution shift, and the objective-level soft NE regularization mentioned in related work.

---

> > > ### Author Response · Authors · 2026-04-03
> > >
> > > We appreciate the reviewer’s follow-up. Our paper takes the value of the NE prior as established (Mohan et al., 2020; Herbreteau et al., 2023) and addresses a different question: how can it be enforced simply? Our iff characterization shows that a parameter-free input-output wrapper recovers the full NE function class on any backbone, including transformers, with no measurable GPU overhead, eliminating the need for constrained architectures.
> > >
> > > The baselines the reviewer suggests, such as multi-noise training, domain adaptation, and related strategies, improve robustness by changing the training distribution or adaptation procedure, rather than by constraining the function class. These are complementary to WNE, not competing alternatives.
> > >
> > > This is why we adopt the single-noise mismatch protocol of Herbreteau et al. (2023), and our results confirm that WNE recovers the structural benefit of the NE prior.
> > >
> > > Regarding the soft NE regularization of Levac et al. (2025): this is the closest conceptual comparator, since it also targets NE. The key distinction is that Levac et al. impose NE through an objective-level soft regularizer, whereas our ideal wrapper enforces input-output NE by construction, with no penalty weight to tune. A direct empirical comparison would be interesting future work, and we note this explicitly in Section 6.

---

### Official Review · Reviewer_jGid · 2026-03-08

**Soundness:** 3
**Presentation:** 3
**Significance:** 2
**Originality:** 2
**Overall Recommendation:** 4
**Confidence:** 3

**Summary:**

This submission proposes a normalisation‑equivariant wrapper (WNE) that enforces robustness to global brightness / contrast via a normalise–process–denormalise factorisation that can wrap arbitrary backbones. The method computes per‑instance mean and standard deviation to project inputs onto a normalised manifold, applies an unconstrained denoiser and analytically restores shift, scale at the output. A formal characterisation proves that this factorisation exactly parameterises the full class of normalisation‑equivariant maps. The crux of the thesis is that modifying internals to maintain equivariance is unnecessary and costly; therefore, enforcing NE at the input–output level yields the same function class and directly improves robustness to contrast / brightness shifts and noise‑level mismatch. Experimental evidence on DnCNN and SwinIR with single‑noise training demonstrates smoother, more stable performance at various test distances from the training noise, alongside additional non‑Gaussian corruption tests. An analytical PSNR decomposition that separates a constant, a normalized‑space regression error term and a scale term depending only on the input standard deviation, may help explain why NE yields stable cross‑noise behavior.

**Compliance With Llm Reviewing Policy:**

Affirmed.

**Final Justification:**

I thank the authors for their rebuttal and link to further explanations and additional experimental work.

The rebuttal serves to clear up some of my initial misunderstandings and adaquelty addresses the majority of my concerns, originally outlined.

**Key Questions For Authors:**

Q. Skip connections enable stable training but can also induce 'lazy training' / shortcut dominance where a residual branch underfits. By analogy, WNE hard-codes a pass‑through of global mean / scale while g operates only on normalised content. Could this structure similarly encourage shortcut dominance or degeneracy? (e.g., $g$ collapsing toward an affine map on $M$, especially at extreme $\sigma$ where $\text{std(y)}$ sets output scale). Can any diagnostics (e.g. variance explained by the pass‑through vs learned path?) increase confidence in the core principle?

Q. The paper primarily reports PSNR. Could you include complementary structure‑aware metrics (e.g., SSIM / FSIM) and error maps across the noise‑mismatch sweeps to verify NE wrapper gains are not accompanied by losses in structural fidelity, that PSNR alone might miss?

Q. Mechanism and inductive bias: Since WNE removes only global mean / scale and reinjects them analytically, to what extent do the observed gains stem from exploiting global statistics versus genuinely improved local processing by $g$ on the normalized manifold $M$? More intuitive checks could include swapping input mean / std between images while holding the normalised content fixed, comparing gains on edges vs flat regions, simple patch-occlusion or shuffle controls, and an identity-backbone ablation to see how much improvement remains without learned local processing.

**Limitations:**

yes

**Strengths And Weaknesses:**

**Strengths**

S1. The challenges being addressed here are real and important; strong and principled solutions that improve robustness to distribution shift in image‑to‑image restoration are of value to the community and have numerous useful applications in practice. The notion of enforcing normalisation equivariance (NE) seems reasonable and well‑motivated as a means to stabilise performance across varying acquisition conditions.

S2. Conceptually somewhat straightforward pipeline. The framework is parameter‑free, backbone‑agnostic, and possible to apply to CNNs and transformers, with experiments reporting positive results and negligible overhead compared to architectural NE designs.

S3. Appropriate analyses and sensitivity studies help reader understanding and isolate contributions of relevant factors, including the characterisation that shows normalise–process–denormalise exactly parameterises NE maps. Writing and paper structure is of a reasonable standard in general. I enjoyed reading the paper.

**Weaknesses**

W1. While the if-and-only-if characterisation is appealing, the practical method closely resembles prior normalise–process–denormalise, adaptive instance normalisation templates (e.g., [a]) and equivariance wrappers, which may temper perceived methodological novelty.

W2. Evaluation scope: main results focus on Set12 / BSD68 greyscale denoising with synthetic corruptions (additive white Gaussian noise plus a few non‑Gaussian / JPEG tests); absence of colour benchmarks, real sensor noise datasets, or larger‑scale modern restoration suites limit claims of broad robustness.

W3. Metrics: Results are primarily PSNR-centric; limited reporting of structural metrics (e.g., SSIM/FSIM), error maps, and perceptual measures makes it difficult to quantitatively assess structural fidelity under mismatch.

W4. Global-statistics assumption: Pooling $\mu$ / $\text{std}$ jointly over all pixels (.. and channels?) enforces a strong global affine model; this may be suboptimal for multi-channel / colour data or images with local illumination / contrast variations. Per-channel or spatially local variants are not investigated.

W5. Experiments beyond denoising problems, diffusion sampler integrations would be welcome and are of interest. Scope limitations are however well noted.

**References**

a. Huang & Belongie. Arbitrary Style Transfer in Real-time with Adaptive Instance Normalization. ICCV 2017.

---

> ### Author Rebuttal · Authors · 2026-03-30
>
> We thank Reviewer jGid for the detailed and thought-provoking review.
>
> **W1: Template resembles prior normalize–process–denormalize work.**
>
> We agree that the template itself appears in prior work (e.g., RevIN for time-series forecasting, AdaIN for style transfer) as an engineering heuristic without connection to equivariance.
> Our contribution is to show that this template is not just one useful trick but is *equivalent* to the full NE function class: Characterization 1 proves that a map is NE *if and only if* it admits the normalize–process–denormalize factorization.
> This changes the template's status from a heuristic to a necessary and sufficient construction.
>
> Concretely, relative to prior NE literature (e.g., Herbreteau et al., 2023), our contributions are:
> (1) the iff characterization itself, absent from RevIN and Herbreteau et al.;
> (2) simplification of architectural NE to a zero-parameter, negligible-overhead, backbone-agnostic wrapper with the same equivariance guarantee—eliminating constrained convolutions, sorting nonlinearities, and the associated $1.6\times$ runtime overhead;
> (3) the $\Delta$-overlap analysis (Section 5), which provides a mechanism explaining *why* NE improves robustness;
> and (4) a new conditional explanation of Observation 1, showing that under a high-dimensional approximation and a contrast-independence assumption, the normalized-space MSE of any estimator is approximately $\Delta$-driven (anonymous link, Section R1).
>
> **W2: Evaluation scope.**
>
> We have broadened the experiments:
> (1) Restormer (transposed attention) with and without WNE under the same mismatch protocol;
> (2) SwinIR non-Gaussian evaluation (Uniform, Laplace, Rayleigh, JPEG);
> (3) color denoising with Restormer on CBSD68.
> All new results are in the anonymous link.
>
> **W3: PSNR-centric metrics.**
>
> We now report SSIM alongside PSNR for all main experiments (DnCNN and SwinIR, Figures 3–4 equivalents).
> The SSIM results confirm that the robustness gains under mismatch are not accompanied by losses in structural fidelity.
> See anonymous link, Figures R2, R3.
>
> **W4: Global-statistics assumption; per-channel variants.**
>
> The color denoising experiment (W2 item 3) tests WNE with statistics pooled jointly over all channels and pixels on 3-channel inputs, confirming that the global pooling strategy works for color data.
> Per-channel or spatially local NE variants are interesting extensions; the characterization naturally generalizes (for any group action, one can ask whether an analogous factorization holds), but we consider these outside the scope of the present submission.
>
> **W5: Beyond denoising.**
>
> We follow the experimental protocol of Mohan et al. (2020) and Herbreteau et al. (2023) for direct comparability.
> Subsequent work has already applied SE/NE denoisers to inverse problems and diffusion samplers (Kadkhodaie & Simoncelli, 2021; Hong et al., 2024; Levac et al., 2025), confirming that the NE prior extends beyond denoising.
> Since the wrapper is parameter-free and backbone-agnostic, it is readily deployable in these settings, though we leave direct validation there to future work.
>
> **Q1: Could the pass-through encourage shortcut dominance?**
>
> The analogy to skip-connection shortcuts does not apply here due to a structural difference in information flow.
> In a residual network, both the skip and the learned branch receive the same input.
> In WNE, the information is *partitioned*: the backbone $g$ receives only $\tilde{y} \in \mathcal{M}$ and has no access to $\mu(y)$ or $\mathrm{std}(y)$.
> With an identity backbone $g = \mathrm{id}$, the wrapped map reduces to $f(y) = \mathrm{std}(y) \cdot \tilde{y} + \mu(y)\mathbf{1} = y$, i.e., the identity map with zero denoising capacity.
> The pass-through channel alone contributes nothing; all denoising must come from $g$ learning a nontrivial map on $\mathcal{M}$.
> Figure 8 confirms that $g$ achieves comparable normalized-space regression quality to the unconstrained Baseline at matched difficulty.
>
> **Q2: SSIM/FSIM.**
>
> Addressed under W3 above.
>
> **Q3: Global statistics vs. learned local processing.**
>
> The wrapper is a parameter-free invertible coordinate transform.
> With $g = \mathrm{id}$, the output is exactly $y$—zero denoising.
> Thus the global statistics channel has no denoising capacity; all gains come from $g$ on the normalized manifold $\mathcal{M}$.
> The wrapper's role is not to process but to *factorize*: it ensures $g$ is queried on normalized inputs, stabilizing the regression problem across noise levels (Section 5).
> Figure 5 provides further evidence that $g$ learns spatially adaptive, edge-aligned filters rather than a trivial map.
>
> Anonymous link: https://anonymous.4open.science/r/rebuttal_icml2026-04FC/rebuttal.pdf

---

> > ### Author Rebuttal · Reviewer_jGid · 2026-04-01
> >
> > I thank the authors for their comprehensive rebuttal and link to further explanations and additional experimental work.
> >
> > The rebuttal serves to clear up some of my initial misunderstandings and adaquelty addresses the concerns originally outlined.

---

### Official Review · Reviewer_bET1 · 2026-03-10

**Soundness:** 3
**Presentation:** 3
**Significance:** 3
**Originality:** 3
**Overall Recommendation:** 3
**Confidence:** 3

**Summary:**

This paper studies a prior property for image-to-image restoration tasks, namely Normalization Equivariance (NE). NE requires that when the input image undergoes a global brightness shift and contrast scaling, the output should exhibit the corresponding shift and scaling as well. The core contribution of the paper is to show that a function satisfies NE if and only if it can be written in the form of
normalize → process → denormalize. Based on this result, the authors propose a very simple wrapper, WNE, which can be directly applied to an arbitrary backbone to enforce NE at the input-output level, without modifying the internal network structure. Experiments are mainly conducted on blind image denoising, and the results show that WNE can significantly improve robustness under noise mismatch while introducing almost no additional computational overhead.

**Compliance With Llm Reviewing Policy:**

Affirmed.

**Key Questions For Authors:**

please see Strengths And Weaknesses

**Limitations:**

please see Strengths And Weaknesses

**Strengths And Weaknesses:**

Strengths
	WNE does not require modifications to the internal layers of the backbone. Therefore, unlike some previous architecture-level equivariant methods, it is not constrained by specific convolutional structures, normalization layers, or attention modules. The paper also demonstrates that it can be applied to different types of networks such as DnCNN and SwinIR.
	Compared with architecture-level methods that require modifications to every layer, this wrapper-based approach is simple to implement and adds little overhead. The experiments also show that WNE introduces almost no noticeable increase in training or inference time on GPU.
	Under the noise mismatch setting in blind denoising, WNE does improve robustness, and its performance is close to that of architecture-level NE methods. This suggests that enforcing NE only at the input-output level can already bring practical benefits.
Weaknesses
	The experimental scope is relatively narrow. Although the theory is presented for arbitrary image-to-image backbones, the experiments are largely concentrated on AWGN blind denoising. As a result, the claimed generality of the method is supported more by theory than by extensive empirical validation.
	The comparisons are not fully controlled in a strict apples-to-apples sense. Some baselines are not evaluated under exactly the same backbone and configuration. For example, the architecture-level NE methods and the standard baselines differ in network details, so these comparisons serve more as contextual references than as fully rigorous direct comparisons.
	The method is not entirely lossless under matched noise conditions. The main advantage of WNE lies in robustness under mismatch, but under matched conditions it is usually only comparable to the baseline rather than consistently better. Thus, what it offers is a better robustness trade-off rather than a strictly free improvement.
	The analysis is more empirically interpretive than theoretically strong. The later discussion on why NE is effective, such as the normalized coordinates and the difficulty variable Δ, is insightful, but it remains more of an empirical explanation than a particularly strong theoretical conclusion.

---

> ### Author Rebuttal · Authors · 2026-03-30
>
> We thank Reviewer bET1 for the thoughtful and constructive review.
>
> **W1: Narrow experimental scope.**
>
> We have substantially broadened the experiments in the revision.
> (1) We add Restormer (Zamir et al., 2022), which uses channel-wise self-attention (transposed attention), trained with and without WNE under the same single-noise mismatch protocol.
> (2) We extend the non-Gaussian evaluation (Uniform, Laplace, Rayleigh, JPEG) from the DnCNN family (Appendix G) to SwinIR Baseline vs. SwinIR-WNE.
> (3) We add color denoising experiments with Restormer on CBSD68, with WNE pooling statistics jointly over all channels and pixels.
> All new results are available at the anonymous link.
> Together with the original experiments, the revised submission covers two CNN backbones (DnCNN, FDnCNN), two transformer backbones (SwinIR, Restormer), four noise types beyond AWGN, and both greyscale and color settings.
>
> **W2: Comparisons not fully controlled.**
>
> We agree that the DnCNN vs. NE-arch comparison in the original submission conflates NE enforcement with architectural differences.
> A fully controlled NE-arch-vs-Baseline comparison is inherently impossible: NE-arch enforces equivariance *by replacing* internal operations (sorting nonlinearities, affine-constrained convolutions satisfying $\mathrm{Conv}_w(\mathbf{1}) = \mathbf{1}$), and the kernel-sum constraint acts as an implicit weight regularizer absent from plain FDnCNN.
> To bracket this confound, we apply WNE to the same FDnCNN backbone used by SE-arch and NE-arch (anonymous link, Table R1 and Figure R1).
> WNE-FDnCNN is competitive with NE-arch across all noise levels (e.g., 34.62 vs. 34.69 at $\sigma=10$, 27.23 vs. 27.26 at $\sigma=50$ on Set12), and mismatch curves follow the same trend as Figure 3.
> The WNE-vs-Baseline comparison, by contrast, is fully controlled: the backbone is identical in both cases, with only the outer wrapper added.
> That WNE-FDnCNN matches NE-arch robustness trends without constrained convolutions or sorting nonlinearities suggests that the robustness gains stem from the NE prior itself, not from the architectural modifications used to enforce it.
>
> **W3: Not lossless under matched noise.**
>
> The matched-noise drop for WNE (e.g., 35.08 $\to$ 34.95 for SwinIR at $\sigma=10$) is small and comparable to the drop seen for NE-arch relative to Baseline in the DnCNN family (Table 1).
> This might reflect the NE constraint provably removing non-NE functions from the hypothesis class—not a deficiency specific to our method.
> We also note that all models are evaluated at the final checkpoint (no early stopping), so part of the small gap may reflect different convergence rates rather than a fundamental capacity loss.
>
> **W4: Observation 1 lacks theoretical justification.**
>
> We have added a conditional explanation for Observation 1, presented in full in anonymous link, Section R1.
> The argument has two parts.
>
> First, at the per-instance level under AWGN, after conditioning on $(x,\tilde{x},\Delta)$, the residual dependence on $\sigma$ disappears exactly: the conditional law of $\tilde{y}$ no longer depends on $\sigma$.
> Geometrically, this is because after conditioning on $\Delta$, the normalized input is distributed on a fixed manifold slice rather than on the full ambient space.
>
> Second, to lift this per-instance statement to a population-level $\Delta$-collapse, we identify one modelling assumption: the normalized image direction $\hat{x} = (x-\mu(x)\mathbf{1})/\|x-\mu(x)\mathbf{1}\|_2$ depends only weakly on empirical contrast $\sigma_x$ over the relevant range.
> Combined with a high-dimensional approximation linking $\tilde{x}$ to $(\hat{x},\Delta)$, this yields an approximate population-level explanation in which the normalized-space MSE depends primarily on $\Delta$, with only weak residual dependence on $\sigma$.
>
> Observation 1 is the empirical counterpart of this prediction: for the trained model, the normalized-space MSE largely collapses as a function of $\Delta$.
>
> We would be grateful if the reviewer could consider revising their assessment in light of these changes.
>
> Anonymous link: https://anonymous.4open.science/r/rebuttal_icml2026-04FC/rebuttal.pdf

---

> > ### Author Rebuttal · Reviewer_bET1 · 2026-04-01
> >
> > I am satisfied that the author has addressed the majority of my concerns. In light of the other reviewers' input and the author's replies, my score remains the same.

---

> > > ### Author Response · Authors · 2026-04-03
> > >
> > > We appreciate the reviewer's follow-up and are glad that the majority of concerns were addressed. As no specific remaining question was raised, we do not have further clarification to add at this stage, but we would be happy to address any concrete residual point during the discussion period.

---

### Official Review · Reviewer_nCHc · 2026-03-11

**Soundness:** 2
**Presentation:** 2
**Significance:** 2
**Originality:** 3
**Overall Recommendation:** 4
**Confidence:** 3

**Summary:**

This paper extends the important characteristic in low-level vision tasks, Normalization Equivariance, to arbitrary network architectures without introducing additional parameters or computational overhead. The theoretical derivation is detailed, but the experimental section is slightly lacking.

**Compliance With Llm Reviewing Policy:**

Affirmed.

**Final Justification:**

The author's detailed supplementary experiments resolved all my concerns. I believe the existing experiments are sufficient to demonstrate the effectiveness of the method proposed in this paper.

**Key Questions For Authors:**

Why not provide experimental results for SwinIR under non-Gaussian noise?

**Limitations:**

1. Insufficient experimentation. It is suggested to provide experimental results with more restoration backbones.
2. It is recommended to comprehensively introduce other equivariance priors.

**Strengths And Weaknesses:**

### Strengths

1. This paper extends the Normalization Equivariance to SwinIR.
2. The proposed methods do not introduce additional parameters or computational overhead.
3. The theoretical derivation is detailed

### Weaknesses

The primary issue with this paper revolves around insufficient experimentation, which prevents a comprehensive verification of the effectiveness of the proposed solution.

1. Although this paper claims that its method applies to arbitrary architectures, the shifted window attention in SwinIR is not a conventional attention method. Conducting experiments only on SwinIR cannot prove the applicability of this method to other attention schemes, such as self-attention [1], sliding window attention [2], linear attention [3], etc.
2. It is noted that the experiments conducted in the maintext are all degraded by Gaussian noise. Although this paper provides experimental results on other non-Gaussian noises in the appendix, the experiments were not conducted on SwinIR. The author should declare this limitation or conduct supplementary experiments.
3. It is recommended to comprehensively introduce other equivariance priors, such as translation [2] and rotation [4], in the "Equivariance priors" section of the "related works" chapter, as they also widely exist in low-level vision tasks.

---

[1] An image is worth 16x16 words: Transformers for image recognition at scale. ICLR 2021.

[2] Enhancing image restoration transformer via adaptive translation equivariance. ICCV 2025.

[3] Mambair: A simple baseline for image restoration with state-space model. ECCV 2024.

[4] Rotation Equivariant Arbitrary-scale Image Super-Resolution. T-PAMI 2025.

---

> ### Author Rebuttal · Authors · 2026-03-30
>
> We thank Reviewer nCHc for the constructive feedback. All three weaknesses are addressed below with new experiments and text changes in the revised manuscript. Figures and tables for the new experiments are available at the anonymous link.
>
> **W1: Applicability beyond SwinIR's shifted-window attention.**
>
> We agree that demonstrating the wrapper on a second attention mechanism strengthens the claim. In the revision, we add Restormer (Zamir et al., 2022), which uses channel-wise self-attention (transposed attention)—a fundamentally different attention scheme from SwinIR's shifted windows. We train Restormer with and without WNE under the same single-noise mismatch protocol ($\sigma_{\mathrm{train}} \in \{10, 25, 50\}$) and report input–output PSNR curves on Set12 (see anonymous link, Figure R4). The results confirm the same pattern: WNE stabilizes performance under mismatch with no architectural modification.
>
> We note that the wrapper is architecture-agnostic *by construction*—it operates entirely outside the backbone via analytic normalization and denormalization (Eq. 8), so its equivariance guarantee holds for any backbone regardless of internal attention mechanism. The Restormer experiment verifies empirically that this theoretical guarantee translates to practical robustness gains on a second, architecturally distinct transformer.
>
> **W2: Non-Gaussian noise experiments not conducted on SwinIR.**
>
> We have now run the non-Gaussian evaluation (Uniform, Laplace, Rayleigh, JPEG) on SwinIR Baseline vs. SwinIR-WNE, using the same input–output PSNR diagnostic as in the main text. Results are reported in anonymous link, Figure R6. WNE improves over the Baseline, consistent with the DnCNN-family results in Appendix G. This addresses the gap the reviewer identified.
>
> **W3: Other equivariance priors in related work.**
>
> We have added a paragraph to the "Equivariance priors" subsection discussing translation equivariance (Hu et al., ICCV 2025) and rotation equivariance (Xie et al., T-PAMI 2025) as complementary spatial equivariance priors in low-level vision. We note that these act on the pixel grid, while normalization equivariance acts on pixel *values* ($y \mapsto ay + b\mathbf{1}$), and that combining spatial and value-space priors is an interesting future direction.
>
> **Key question: Why not provide SwinIR results under non-Gaussian noise?**
>
> This is now addressed by the new experiments described under W2 above.
>
> We hope these additions—a new backbone (Restormer), SwinIR non-Gaussian results, and the expanded related work—address the reviewer's concerns. We would be grateful if the reviewer could consider revising their assessment in light of these changes.
>
> Anonymous link: https://anonymous.4open.science/r/rebuttal_icml2026-04FC/rebuttal.pdf

---

> > ### Author Rebuttal · Reviewer_nCHc · 2026-04-02
> >
> > The author's detailed supplementary experiments resolved all my concerns. I raised my score to weak accept.

---

### Decision · Program_Chairs · 2026-04-30

**Decision:**

Accept (regular)

**Comment:**

This paper generalizes the key property of normalization equivariance in low-level vision to arbitrary network architectures, achieving this without introducing additional parameters or computational overhead. Reviewers appreciated the importance of the problem, the architecture-agnostic nature of the approach, and the mathematically rigorous insights into equivariant function structures.

The primary concern was the limited scope of the experimental evaluation, with suggestions to include a broader range of attention mechanisms, noise types, datasets, and metrics. The rebuttal substantially expanded the experiments, and all reviewers acknowledged that these additions effectively addressed their concerns.

A remaining discussion point is the comparison with alternative strategies such as multi-noise training and domain adaptation. The AC agrees with the authors that these approaches are complementary to the proposed method.

As the major concerns have been adequately resolved, and given the paper’s strengths, including its formulation of an important problem, strong theoretical grounding, and practical advantage of introducing no additional parameters or computational overhead, the AC recommends acceptance.